# B'MOJO: Hybrid State Space Realizations of Foundation Models with Eidetic and Fading Memory

**Luca Zancato**[*]    **Arjun Seshadri**    **Yonatan Dukler**    **Aditya Golatkar**    **Yantao Shen**

**Benjamin Bowman**    **Matthew Trager**    **Alessandro Achille**    **Stefano Soatto**

**AWS AI Labs**

## Abstract

We describe a family of architectures to support transductive inference by allowing memory to grow to a finite but a-priori unknown bound while making efficient use of finite resources for inference. Current architectures use such resources to represent data either eidetically over a finite span ("context" in Transformers), or fading over an infinite span (in State Space Models, or SSMs). Recent hybrid architectures have combined eidetic and fading memory, but with limitations that do not allow the designer or the learning process to seamlessly modulate the two, nor to extend the eidetic memory span. We leverage ideas from Stochastic Realization Theory to develop a class of models called B'MOJO to seamlessly combine eidetic and fading memory within an elementary composable module. The overall architecture can be used to implement models that can access short-term eidetic memory "in-context," permanent structural memory "in-weights," fading memory "in-state," and long-term eidetic memory "in-storage" by natively incorporating retrieval from an asynchronously updated memory. We show that Transformers, existing SSMs such as Mamba, and hybrid architectures such as Jamba are special cases of B'MOJO and describe a basic implementation that can be stacked and scaled efficiently in hardware. We test B'MOJO on transductive inference tasks, such as associative recall, where it outperforms existing SSMs and Hybrid models; as a baseline, we test ordinary language modeling where B'MOJO achieves perplexity comparable to similarly-sized Transformers and SSMs up to 1.4B parameters, while being up to 10% faster to train. Finally, we test whether models trained inductively on a-priori bounded sequences (up to 8K tokens) can still perform transductive inference on sequences many-fold longer. B'MOJO's ability to modulate eidetic and fading memory results in better inference on longer sequences tested up to 32K tokens, four-fold the length of the longest sequences seen during training.

## 1 Introduction

In Machine Learning, data *representations* are parametric maps trained to *re-present* data either individually, by optimizing a reconstruction criterion, or collectively, by optimizing a classification criterion. A trained representation can be co-opted to map a previously unseen datum to a hypothesis, for instance a class label. Representation learning with deep neural networks has been at the core of recent progress in artificial intelligence (AI) during the past decade. This paper is instead concerned with data *realizations*, which are maps trained to *realize*, *i.e.,* to make real, bring into existence, or *generate* data. Roughly speaking, realizations are "generative representations," trained by optimizing a prediction criterion, that can be used as sequential decision or prediction maps, or as generative

---

[*]Correspondence to: zancato@amazon.com

38th Conference on Neural Information Processing Systems (NeurIPS 2024).

models for sequence data. Large language models (LLMs) and other predictive models that involve randomness in the sequential generation process are special cases of *stochastic realizations* [28].

Representations are the backbone of *inference from inductive learning*, or induction for short. Induction refers to the process of mapping properties of a *particular* set of training data, through the parameters of the learned map, to properties of *general* test data. Successful induction, leading to *generalization*, hinges on an assumption of stationarity, namely the existence of some unknown distribution from which both past (training) data and present (inference) data are independently and identically drawn (IID). While uniform generalization bounds provide provable guarantees for any distribution, they do so under the assumption that the distribution exists and is the *same* for training and testing. This assumption is routinely violated in practice (*e.g.*, language, climate, and business data), as manifest in so-called "out-of-distribution" effects or "distribution shift." These lead to apparent paradoxes involving generalization and memorization [55].

Realizations, on the other hand, are the backbone of *transductive inference* and generative AI (GenAI). Transduction refers to the process of inferring *particular* properties of test data by processing, at inference time, all given (*particular*) training data.[2] The boundary between induction and transduction is blurry: Induction can be viewed as a restricted form of transduction, where training data is accessible only through the learned weights. An over-parametrized representation, when overfit to training data, could in principle store the training set in the weights, thus making induction functionally identical to transduction. However, *optimal induction* aims to foster generalization by using various forms of regularization to prevent memorization, leading to *sub-optimal transduction*. Another form of sub-optimal transductive inference is termed "in-context learning" – which notably involves no learning if by learning one means to "improve by experience:" The same in-context task, presented multiple times, requires identical effort and leads to no improvement. In-context learning can be optimal transduction only if all the data of interest fits in the context (including the entire training set), and even then it has been proven optimal for Transformers only for simple tasks such as linear classification. In summary, memorization and specific inference computation at the core of transduction are in contrast with the biases fostered by inductive learning: Whereas inductive inference seeks to minimize memorization to avoid overfitting and to foster generalization to unseen data, transductive inference seeks to *maximize memorization* and *forgo generalization in favor of sample-specific inference computation*.[3]

Transduction does not require the train and test distribution to be the same. Indeed, the joint distribution from which both present and past data could have been jointly drawn can change with every sample. Therefore, optimality is not measured relative to *one unknown* distribution, as in generalization bounds, but rather relative to *all possible* distributions. If the data is generated by a physically realizable process, such distributions are *computable*. Optimal inference measured on average over all possible computable distributions, weighted by the Universal Prior, has been described by Solomonoff [41]. Solomonoff-style inference can be thought of as the limit of transduction, and similarly involves no learning. Instead, it consists of cycling through all computable programs using a Universal Turing Machine, which requires infinite time, memory, and compute resources, which in turn renders such inference unattainable.

Nonetheless, this Solomonoff limit points to two directions for improving inference: (a) efficient memorization, ideally by losslessly encoding all past data, and (b) efficient test-time computation through hardware and model co-design. Ideally, the resulting realizations would be such that, if memory and compute resources were extrapolated to infinity, inference would approach the Solomonoff limit. In reality, inference is always resource bound, so if something has to grow it would have to be external to the core inference engine. Accordingly, our goal in this work is to design and analyze families of architectures that (a) natively incorporate retrieval from a growing external memory (retrieval-augmented generation, or RAG), and (b) can scale to perform efficient inference.

In order to design families of architectures that efficiently memorize and scale inference computation, we must exploit the structure of the data they realize. When past and present data are known to

---

[2]Sometimes ambiguously termed 'test-time computation' [7, 44], which is confusing since inductive inference also involves test-time computation. Another term sometimes used is 'test-time training' [42, 51], which is also confusing since 'training' and 'testing' are disjoint and complementary phases of inference from inductive learning. All these are different forms of transduction.

[3]We report more examples on the differences between representations/realizations and induction/transduction in Appendix A.

be independent and identically distributed (IID) one can encode inference computation through a fixed "stateless" map, or *model*, which is a representation that can be trained inductively regardless of the inference query. When past and future data are not IID but generated by a Markov process of *known* order, there exist finite-dimensional statistics, called *states*, that summarize all past data for the purpose of prediction [25, 22, 28]. But even if the underlying process is Markovian of bounded order, unless such an order is *known* a-priori optimal realization is generally not possible with constant complexity [32]. To perform optimal inference, memory has to grow, and if the data generation mechanism has finite complexity at some point an efficient encoding of past data into memory will stop growing, but it is not possible to know when [1]. Therefore, a suitable architecture has to always allow the possibility of adding new storage.

In the seventies, Stochastic Realization Theory [11] studied State Space Models (SSMs) under the known-order Markov assumption, since realizations with growing memory were unmanageable then. [4] Today, Foundation Models can ingest a large portion of the growing volume of data accessible through the Internet and make it available for inference. Current AI systems are typically hard-constrained by inference time and compute resources, but not by storage—one can always add more disks. To extend Stochastic Realization beyond the IID or known-Markov cases, Foundation Models need scalable architectures that comprise short-term memory updated synchronously within the given computational constraints and long-term memory updated and accessed sparingly and asynchronously. The former includes both eidetic (lossless) and fading (lossy) memory for efficient computation, while the latter is akin to an integrated form of "retrieval-augmented" inference. Such architectures would seamlessly manage short-term eidetic memory "in-context", fading memory "in-state", long-term structural memory "in-weights" and long-term eidetic memory 'in-storage' [13].

Existing architectures such as Transformers and SSMs fall short of encompassing these criteria. Transformer-based architectures use eidetic memory restricted to a finite span, "context length" [6, 23], while recent SSM-based architectures [18, 16, 49] use only fading memory in their state. In both cases, scaling requires allowing the context (and the key-value cache) or recurrent state to grow unbounded. Recent work on hybrid combinations of Transformer and State Space layers [27, 38, 10] show promise in striking a balance between eidetic memory, fading memory and compute.

## 1.1 Contributions

In this work, we describe a class of models that encompasses both recent SSMs [18], Transformers [23], and hybrid architectures [27, 10] as special cases, which we call B'MOJO. This model family simultaneously renders the high expressivity and recall of Transformers, and the high compute efficiency of SSMs. And, rather than assigning tokens to the attention mechanism by recency, an asynchronous selection mechanism assigns tokens based on unpredictability. That is, whenever the model processes a new token that cannot be well-explained it will append it to the registers that implement B'MOJO's eidetic memory—a process we call Innovation Selection.

We demonstrate through synthetic tasks that B'MOJO outperforms existing SSM and hybrid model architectures in transductive inference. Empirically, we show that our implementation is 15% faster than similarly-sized Transformers and SSMs while achieving comparable perplexity on language model tasks up to the 1.4B scale. Finally, we show that B'MOJO can operate effectively at inference time on sequences far longer (tested up to $4\times$) than those used for training. Specifically, experiments with a B'MOJO architecture trained with sequences of at most 8K tokens show consistent length generalization on test sequences of 32K tokens.

## 2 Background and Related Work

We start with a general overview of models for sequence data, whether of text tokens (Large Language Models), images (*e.g.*, Vision Language Models), video, or other physical sensory data (World Models). *Any* predictive model inferred from a sequence is called a *realization*.[5] If one assumes that there exists *one* "true" model, coming from a known model class, then the problem of realization reduces to System Identification [30], which is to estimate the true model parameters from data. If the model is linear and driven by Gaussian noise, the Cayley-Hamilton theorem [24] implies that

---

[4]Under the stationary assumption, one stochastic realization of an SSM is a convolution. This input-output map is equivalent to an SSM and standard transformations can be applied from one to the other.

[5]Diffusion Models, a special case, are treated as realizations during training but used as representations during inference since all but the final element of the denoised sequence are discarded.

the model's "true" order can be inferred with a variety of model selection criteria, including greedy selection. Otherwise, the rank of the non-linear analog of the observability matrix, the Observability Codistribution, can grow in fits and starts, making greedy model selection undecidable [21]. If not only the model class, but the model itself are known, then the problem further reduces to *filtering* or *prediction*, which for the linear-Gaussian case can be implemented as a closed-form iterative update [25]. When building a (generally non-unique) realization, all one is given is the data, which leaves complete freedom of choice of model class and order, or number of free parameters.

**Stochastic Realization.**   A "sequence model" is a mathematical representation of sequential data capable of predicting the next datum in the sequence given previous ones. The Wiener Filter [46] was among the earliest to be deployed, superseded by the Kalman Filter [25], the first State Space Model (SSM) with an explicit "state" updated recursively. The ensuing decades saw extensions to more general models leading to Stochastic Realization Theory [28]. The general problem of stochastic realization is, given a (potentially infinite) sequence $\{\ldots, u_{t-1}, u_t\} \doteq u_{\leq t}$ observed up to time $t$, infer (*i.e.*, "learn") some parametric model, a function $\phi$ with parameters $\theta$, $\phi_\theta(u_{\leq t})$ such that the prediction $\hat{u}_{t+1} = \phi_\theta(u_{\leq t})$ yields a residual $\epsilon_{t+1} = u_{t+1} - \hat{u}_{t+1}$ ("innovation process") that is as close as possible to independent and identically distributed samples (IID). In a nutshell, *an optimal predictor is one that makes the prediction error unpredictable.*

**State.**   The state of a *model* is *a statistic that makes the future independent of the past.* In particular, such a statistic (function of past data) $\xi(u_{\leq t})$ yields to the following conditional entropy equality $H(u_{t+1}|\xi(u_{\leq t}), u_{\leq t}) = H(u_{t+1}|\xi(u_{\leq t}))$ relative to the joint distribution of all past and present data [30, 29]. Trivially, the data itself fits the definition of state with $\xi(u_{\leq t}) = u_{\leq t}$, but it grows unbounded over time. Instead, one seeks a *bounded complexity* state given which all past data can be ignored with no loss of information. Building such a state is the core goal of stochastic realization.

**Elementary model classes: LTI, LTV and LIV.**   Any finite sequence can be realized by a linear time-invariant (LTI) system driven by IID Gaussian noise [28]. It would therefore appear that this model class is sufficient for any practical purpose, and we need models no more expressive than linear time-invariant ones driven by white zero-mean Gaussian noise:[6]

$$\begin{cases} x_{t+1} = Ax_t + Bu_t \\ y_t = Cx_t + v_t \end{cases} \quad \text{LTI} \qquad \begin{cases} x_{t+1} = A(u_t)x_t + B(u_t)u_t \\ y_t = C(u_t)x_t + v_t \end{cases} \quad \text{LIV}$$

Given observations $y_{0:t}$, stochastic realization deals with the problem of inferring an equivalence class of model parameters $A, B, C$ [43] and a state $x_t$ along with a covariance matrix $P_t$ of $[u_t, v_t]$ that can be propagated deterministically to approximate trajectories produced by the underlying data generation mechanism, $y_{0:t}$ [47]. However, arbitrarily long and complex time series would require a growing state, therefore making the model no longer time-invariant. More expressive model classes, such as time-varying [22] and input-varying [26] afford more efficient representation by considering $A_t, B_t, C_t$ *known* functions of time (Linear-Time Varying) or of their *input* (Linear-Input Varying, LIV). As we describe next, a special case of LIV model class where the dependency on the input is linear, resulting in a so-called *bilinear realization*, is gathering considerable attention [18, 10, 49].

**Modern realizations.**   Input-dependent (bilinear) models first gained popularity half a century ago starting with Brockett [5], when [26] showed that *"every nonlinear system with controls entering linearly is locally almost bilinear"*; [11] developed a complete realization theory for this class of systems including *minimal realizations*, and showed that they not only minimize the dimension of the state, but also *minimize the number of operations necessary for the update.* Most recently, special cases of bilinear realizations are being used as building blocks in stacked architectures like Mamba [18], Jamba [27] and Griffin [10]; they refer to input-dependency as "selectivity" [18, 19, 36] and combine Attention mechanisms and other techniques to scale up to 52B parameters. Similarly, but at a smaller scale, Block State Transformers [14], GSS [33] and H3 [16] consider using State Space Models to contextualize information that is later aggregated with Attention. All these models differ by their choices of $A, B, C$, each with its own advantages and limitations which we describe next.

---

[6]Note that Stochastic Realization Theory does not *assume* that the *data* is Gaussian, zero-mean and white, it only *requires* the *innovation process* to be so. That is the condition that the prediction residual (innovation) be unpredictable, regardless of whatever distribution the data is drawn from.

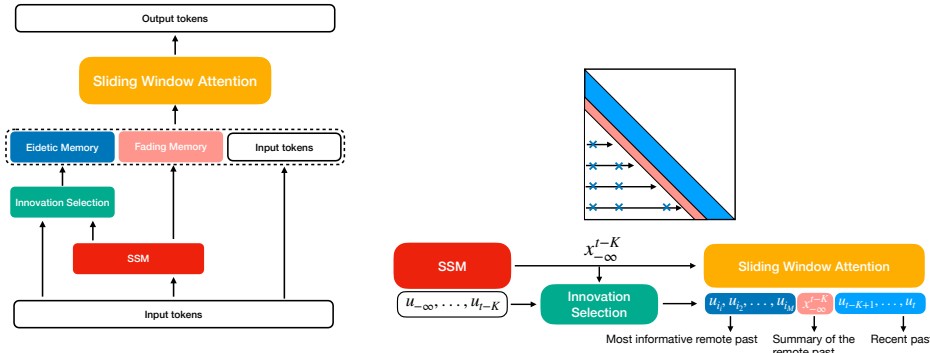

Figure 1: **B'MOJO's memory management. (Left) Illustration of the B'MOJO layer. (Right) B'MOJO's Realization.** B'MOJO's fading memory is computed by a SSM that represents long-range dependencies through its state (a fixed-dimensional representation) which is later aggregated along with with the most recent past. B'MOJO's eidetic memory stores tokens selected from the past using an innovation test on the SSM's state and appends them to the current sliding window. The innovation test measures how difficult it is to predict the next token using the SSM's state. If a token is difficult to predict, we store it in the eidetic memory and pass it to the attention module together with the state, a compressed summary of the past, and the most recent tokens.

## 3 B'MOJO

We now introduce B'MOJO, a general family of architectures based on a stackable module, designed to foster transductive inference. We represent *fading memory* using the state of a dynamical model whose size is fixed a-priori based on hardware constraints. Since the state of a dynamical model is a lossy memory of past data, we implement a complementary *eidetic memory* with shifting registers that directly encode past data as new information is processed. Although adding storage is simple and cheap, peering back into the growing history with every new query is not. Approximate retrieval [34, 50] also grows as more tokens are processed, potentially sub-linearly if optimized off-line. We propose to access information arbitrarily far in the past at a fixed compute cost by keeping only the most unpredictable tokens according to an innovation test. While we respect hard constraints on the *amount* of memory processed at inference-time, we impose no constraint on its time *span*.

### 3.1 Fading Attention

We begin by describing the elementary components of B'MOJO, which we show to encompass existing models. We first show that the Attention mechanism implements a non-linear nil-potent system whose state grows as more samples are processed. Much like a Moving-Average (MA) system with non-linear read-out, the attention state is required to increase as its span is enlarged. Then, we show that Mamba [18] has a fixed-dimensional Auto-Regressive (AR) (fading) state and hence cannot perform exact recall. Finally, we describe a model that modulates the two forms of memory, B'MOJO-Fading (B'MOJO-F), effectively realizing a non-linear ARMA model [30, 52, 53].

**Transformers** use Attention to map input to output sequences according to $y_t = \frac{\sum_{i=1}^{t} \exp(q_t^T k_i) v_i}{\sum_{i=1}^{t} \exp(q_t k_i^T)}$, where $q_t, k_t, v_t$ are the query, key and value vectors and are all computed directly from the input tokens $u_t$. We write this equation as a nil-potent dynamical system with a softmax read-out function $\rho$ as follows:

$$x_{t+1} = A(u_t)x_t + B(u_t); \qquad y_t = \rho(u_t, x_t) \tag{1}$$

where

$$A(u_t) = A_{\text{ATT}} = \begin{bmatrix} 0 & I & & \\ & & \ddots & \\ & & & I \\ & & & 0 \end{bmatrix} \in \mathbb{R}^{2VN \times 2VN} \quad \text{and} \quad B(u_t) = b_{\text{ATT}}(u_t) = \begin{bmatrix} 0 \\ \vdots \\ 0 \\ b(u_t) \end{bmatrix}$$

with $b(u_t) := \begin{bmatrix} k_t \\ v_t \end{bmatrix} \in \mathbb{R}^{2V}$, where $2V$ is the embedding dimension of the KV cache and $N$ is the length of the Attention window. A Transformer has *only short-term eidetic memory* that is deadbeat in $N$ steps: what slides "out of context" is permanently removed.

**Mamba** is complementary in that it *only has fading memory* [18] with decoupled (diagonal) dynamics. Specifically,

$$A(u_t) = A_{\text{Mamba}}(u_t) = \begin{bmatrix} a_1(u_t) & & \\ & \ddots & \\ & & a_N(u_t) \end{bmatrix}, \quad b(u_t) = b_{\text{Mamba}}(u_t) = \begin{bmatrix} b_1(u_t) \\ \vdots \\ b_N(u_t) \end{bmatrix}$$

where $N$ is picked a-priori and the output is $y_t = C(u_t)x_t + Du_t$. A Mamba model is obtained by stacking diagonal Mamba layers, hence insufficient to realize even a simple oscillator without resorting to (numerically slow) complex algebra. A marginally stable pole ($a_i = 1$) can be used as permanent memory but at an unbounded cost of encumbering the state and re-processing it at each time step. [18] emphasizes the use of "selective state space," which is simply a bilinear realization [12, 11]. However, crucial to its implementation are interleaved convolutional layers that retrieve some of the eidetic functionality lost with the diagonal dynamics. In Sect. E we derive in detail the form above from the description of the paper, forgoing the unnecessary continuous time narrative, and comment on the model in relation to the actual implementation.

**B'MOJO-F** bypasses the limitations of Transformer and Mamba layers by using a Controllable Canonical Form [24] to realize Equation (1) (see appendix C), that is:

$$A_{\text{B'MOJO}}(u_t) = \begin{bmatrix} 0 & I & & \\ & & \ddots & \\ & & & I \\ a_1(u_t) & a_2(u_t) & \dots & a_N(u_t) \end{bmatrix}, \quad b_{\text{B'MOJO}}(u_t) = \begin{bmatrix} 0 \\ \vdots \\ 0 \\ b(u_t) \end{bmatrix}$$

In Appendix C we show that B'MOJO-F is a minimal realization and, when the state dimension is fixed, it generalizes both Mamba and Attention modules. B'MOJO-F uses a fixed computational budget to capture long-range dependencies far beyond the span of Attention thanks to the last rows in the state transition matrix. Specifically, the order $N$ fixes the number of tokens (the most recent ones) that are processed by a local attention window $\rho$ thanks to the upper diagonal identities in $A_{\text{B'MOJO}}(u_t)$. On the other hand, the last rows of $A_{\text{B'MOJO}}(u_t)$ aggregate information from the most recent past data, akin to "attention sinks" [48].

### 3.2 B'MOJO's complete functional form

While B'MOJO-F generalizes both Transformers and Mamba models, it can only access information outside the current window with lossy fading memory. Data that becomes relevant only after a long time span would be ignored. We therefore modify our model class to detect and incorporate such tokens into the span of the local attention *eidetically*. We do so with a simple method that we call *Innovation Selection*: whenever B'MOJO processes a new token that cannot be explained using the lossy fading memory, we affix it to the eidetic memory. Innovation Selection operates similarly to the mechanism behind the Lempel-Ziv-Welch (LZW) compression algorithm [56, 45].[7] Like with fading memory, the tokens in the new eidetic memory are processed by the same sliding window attention we use in B'MOJO-F. Algorithm 1 captures the steps we perform from input to output of each layer of B'MOJO architecture, which we discuss below.

---

**Algorithm 1:** The B'MOJO Mechanism

---

**Data:** Input data $u_t$, inner recurrent state $x_{t-1}$, fading output $y_{t-1}$, eidetic memory $M_{t-1}$, window size $w$, B'MOJO's state update matrix $A(u_t)$, input to state vector $B(u_t)$, state to output vector $C(u_t)$, a predictor function $\hat{y}(\cdot)$ over a span of length $k$.

**Result:** Output $\text{out}_t$

$u_{t-w:t} \leftarrow u_{t-w-1:t-1} \cup u_t$ ;      // Short-term memory window

$x_t \leftarrow A(u_t)x_{t-1} + b(u_t)$ ;    $y_t \leftarrow C(u_t)x_t$ ;      // Long-term fading memory

$\epsilon_t \leftarrow \text{error}(\hat{y}(y_{t-1:t-k}), y_t)$ ;      // Innovation computation

$M_t \leftarrow \begin{cases} M_{t-1} \cup \{u_t, \epsilon_t\} & \text{if } \epsilon_t > \min_{\epsilon \in M_{t-1}}(\epsilon) \\ M_{t-1} & \text{otherwise} \end{cases}$ ;      // Long-term eidetic memory

$\text{out}_t \leftarrow \text{Attn}(u_{t-w:t}, y_{t-w}, M_t)$ ;      // Model output

---

[7]In brief, LZW scans an input sequence to find the shortest sequence that is currently unknown, appends this sequence to a dictionary while returning indices for known sub-sequences. Similarly, Innovation Selection scans the input sequence to find the shortest sequence that has high prediction error, adds to the eidetic memory unpredictable inputs, while returning outputs that leverage known sub-sequences.

In Algorithm 1, the first line updates the short-term memory (collection of the last $w$ seen tokens) by ejecting the oldest token and appending the new token. The second line updates the SSM fading memory, which stores new information into the state $x_t$. Then, we augment fading memory with our *eidetic* memory $M_t$, a set of tokens that the model decides to keep based on their unpredictability: tokens that are difficult to predict given the past, as measured by $\text{error}(\hat{y}_t(y_{t-1:t-k}), y_t)$, could be valuable far in the future when their memory has faded away, and are hence curated in the *eidetic* memory. The amount of information stored in $M_t$ can grow unbounded over time up to hardware limitations, in Section 3.3 we further discuss how to efficiently implement the predictor $\hat{y}$.

Finally, we weave fading and eidetic memory together to predict a new token through a sliding attention mechanism, which aggregates relevant information from short-term memory $u_{t-w:t}$, the fading memory $y_t$ and the long term eidetic memory $M_t$.

**Differences between B'MOJO, B'MOJO-F and vanilla Hybrid models**. Differently from vanilla hybrid models that stack SSM and Attention layers, B'MOJO-F and B'MOJO allows the Attention module to attend to both the input and output tokens of the SSM, allowing the Attention layer to merge information from fading "memory" tokens (output of the SSM) with "eidetic" tokens (the layers' inputs). Differently from B'MOJO, B'MOJO-F does not use the innovation selection mechanism and therefore it does not implement long term eidetic memory.

### 3.3   B'MOJO's efficient implementation

We efficiently implement B'MOJO's tiered memory hierarchy at scale. The modelling choices that lead to B'MOJO closely follow ideas from Stochastic Realization and have an obvious recurrent efficient implementation. On the other hand, during training, one is more interested in a parallel formulation that can allow processing of multiple tokens at the same time and in a single forward pass. In the following, we describe how we use chunking to develop B'MOJO's parallel form.

**Efficient sliding window and memory chunking.** Computing fading memory and selecting the most unpredictable tokens to store before feeding them into an attention layer is a sequential process which is hard to parallelize. To solve this we use chunks of length $w$ and use fading and eidetic memory to summarize the whole past before the beginning of each chunk. Then, to aggregate information from the input and memory tokens, we use a sliding window over the interleaved concatenation of input chunks and their respective memory tokens as shown in Figure 7 (see Appendix B.2). Our modular and interleaved approach in Figure 7 enables us to leverage optimized kernels for both SSM [18] and sliding window attention [9], enabling fast training and inference beyond the 1B scale. In Figure 4 we show that our efficient implementation is faster than both Mamba [18] and Mistral [23].

**Efficient Innovation Selection.** The Innovation Selection process we describe in Algorithm 1 requires the predictor $\hat{y}(y_{t-1:t-k})$. While building a new parametric predictor with learnable parameters is possible, in practice, this requires modifying the training loss. In our implementation, we consider a fixed predictor so no extra weights need to be learned. We fix $\hat{y}$ as a running weighted average and implement it using short 1D grouped causal convolutions.

## 4   Experimental Results

To evaluate the scalability and efficiency of B'MOJO, we compare it with state-of-the-art model classes (SSMs, Transformers and hybrid variants) on both synthetic and language modeling tasks. Our experimental results are of two main types, (i) language modeling scaling laws (Section 4.2) with zero-shot evaluation on short context text benchmarks (Section 4.3), and (ii) specific long context/recall-based tasks where finite-context models are ill-fit. We wish to emphasize that in this setting, and for the results of type (i), Transformers are a paragon, not a baseline, since most tasks are answerable within the context. Therefore in the results of type (ii) we leverage specific benchmarks, like synthetic tasks (Section 4.1), long context evaluation and length generalization to assess our models (Section 4.3). The goal of our novel model class is to cover the entire spectrum, i.e. perform comparably to the paragon on finite contexts while preserving higher performance than Transformers whenever the data relevant to solve the inference tasks fall outside the context window.

All experiments compare against three baselines: (1) Transformers, represented by a downscaled Mistral-7B [23] architecture re-trained from scratch (2) SSMs, represented by Mamba and (3) Hybrid models, implemented by stacking Mamba with a sliding window attention [10, 27]. For a fair apples-to-apples comparison all our models are trained from scratch using the same pre-training data,

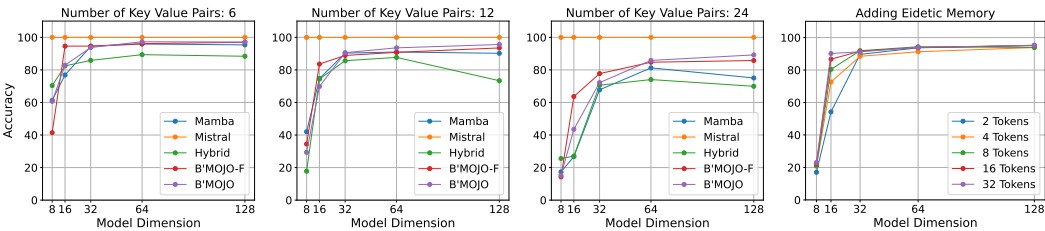

Figure 2: **(Panels 1-3) B'MOJO has high memory efficiency on Associative Recall Tasks (sequence length is 256 and attention window 32).** For various models, we plot accuracy on the Multi-Query Associative Recall (MQAR) task as a function of the model dimension (totaling the SSM state, eidetic memory and KV cache where applicable). The transformer paragon attains 100% accuracy because it operates on the full context. While all models benefit strongly from increased memory, B'MOJO and B'MOJO-F consistently achieve the best accuracies for a given memory budget. Panels 1-3 report MQAR tasks of increasing difficulty, on which the performance gap between B'MOJO and other models increases, showcasing the value of eidetic memory. **(Panel 4) Increases in eidetic memory size corresponds to gains in recall**. We ablate the effects of eidetic memory by growing the number of eidetic memory tokens in B'MOJO. Each added token contributes to an increase in recall accuracy until performance is saturated.

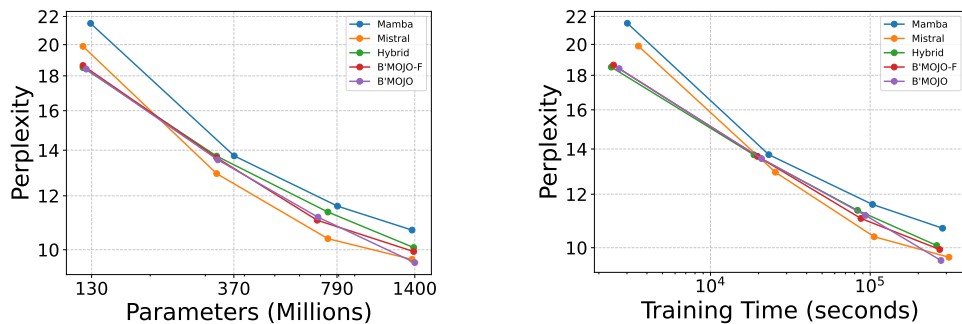

Figure 3: **B'MOJO language modeling scaling laws.** We plot the perplexity reached by models at different scales against the number of parameters and the wall-clock training time. B'MOJO is faster than Mamba and Mistral at training time while achieving better perplexity than Mamba and comparable perplexity with Mistral. The plot also exhibits a non-saturating scaling law, showing that increasing the amount of resources leads to increasingly better B'MOJO models.

tokenizer [23], and context length. And, in order to abalte the contribution of eidetic memory, we consider B'MOJO-F in addition to B'MOJO in all our experiments.

## 4.1 Synthetic Tasks

We use synthetic tasks to test B'MOJO's ability to recall exact information from beyond the attention span [2, 49]. We do so with Multi-Query Associative Recall (MQAR) data in the main text, and consider other tasks such as Induction Heads [35], Selective Copying [18], Fuzzy MQAR[38], and Noisy MQAR [38] in Appendix D.1.

**B'MOJO's Memory Efficiency on Associative Recall Tasks.** The MQAR task [2] has been shown to correlate well with language modeling performance [2, 38]. Compared to its peers (e.g. Induction Heads [35]) MQAR is considerably more difficult and requires strong recall capabilities. In Figure 2, we display accuracy on MQAR as we vary the size of the recurrent state for 2-layer instances of our models. Panels 1-3 consider varying numbers of key-value pairs to illustrate increasing complexity. Here, a Transformer (Mistral), with its sequence-length-sized KV cache serves as the paragon, always achieving a 100% accuracy. Should its window size be restricted not to include the KV pairs, its accuracy would drop to that of a random guess. Our results show that while every model class improves in recall accuracy as its size increases, B'MOJO and B'MOJO-F do so more reliably and faster than others. We explain these findings as follows. Mamba possesses only fading memory to propagate information to the future, and therefore has a limited recall capacity [2] for a given size of its recurrent state. Hybrid models leverage a sliding window attention to mitigate this issue,

Table 1: **B'MOJO's performance on downstream tasks.** We compare different architectures on several zero-shot downstream tasks used to test common-sense reasoning and question-answering on relatively small contexts. These tasks, however, do not require strong recall capabilities because the input text is typically very short (results on longer contexts are reported in Table 2). On pre-training perplexity B'MOJO performs on par with our pre-trained Mistral model and outperforms our pre-trained Mamba models at the largest scale we test 1.4B. However, on accuracy metrics, while B'MOJO still outperforms Mamba, its gap with the Mistral model increases.

| | | Pre-training Log-Perplexity | Short Context (acc ↑) | | | | | | |
| | | | LAMBADA [37] | HellaSwag [54] | PIQA [4] | ARC-E [8] | ARC-C [8] | WinoGrande [39] | Avg. |
|---|---|---|---|---|---|---|---|---|---|
| 370M | Mistral (Full-Attention) | 2.56 | 31.6 | 33.8 | 64.0 | 44.9 | 23.5 | 50.4 | 41.4 |
| | Mamba (SSM) | 2.62 | 31.4 | 33.4 | 63.5 | 45.0 | 22.3 | 51.7 | 41.2 |
| | Hybrid (Sliding Attention + SSM) | 2.69 | 26.3 | 31.3 | 61.1 | 42.7 | 22.4 | 51.9 | 39.3 |
| | BMoJo (Fading) | 2.68 | 29.6 | 33.2 | 63.7 | 43.1 | 23.0 | 51.8 | 40.7 |
| | BMoJo (Fading + Eidetic) | 2.67 | 28.6 | 33.3 | 63.9 | 44.3 | 22.1 | 50.7 | 40.5 |
| 1.4B | Mistral (Full-Attention) | 2.27 | 50.1 | 50.7 | 70.4 | 58.2 | 27.5 | 54.4 | 51.9 |
| | Mamba (SSM) | 2.37 | 43.9 | 45.0 | 70.3 | 52.4 | 28.0 | 51.9 | 48.6 |
| | Hybrid (Sliding Attention + SSM) | 2.42 | 37.6 | 38.8 | 66.1 | 48.4 | 25.4 | 52.6 | 44.8 |
| | BMoJo (Fading) | 2.27 | 45.4 | 46.0 | 70.0 | 52.3 | 26.6 | 53.3 | 48.9 |
| | BMoJo (Fading + Eidetic) | 2.26 | 44.8 | 46.8 | 69.9 | 54.7 | 26.6 | 52.1 | 49.1 |

however they require a lengthy window to increase their span and recall the reference pairs. The strong performance of B'MOJO-F showcases the value of fading memory over a simple hybrid configuration, and the even stronger performance of B'MOJO, most evident in the panel on the right, highlights the added contribution of an eidetic memory for precise recall.

## 4.2 Language Modeling Scaling laws

We next demonstrate B'MOJO's favourable scaling laws on mid-size language modeling, baselining against the same set of state-of-the art-model classes as the previous section. We report the training setting and hyper-parameters in Appendix B.1.

**Results.** In Figure 3, we report perplexity at different scales against (left) the number of parameters and (right) the wall-clock training time. B'MOJO is faster than Mamba and Mistral that use efficient CUDA kernels (Flash attention and Selective Scan), outperforms our pre-trained Mamba baseline at all scales, and is comparable with our Mistral Transformer model, in Appendix D.2 we further comment our results in relation to other recent hybrid models like Griffin [10].

We additionally report the scaling behavior of B'MOJO-F as an ablation of the eidetic memory, as well as a vanilla hybrid architecture composed of interleaved SSM and attention layers as an ablation of both fading and eidetic memory. Despite using only short context sizes (2k) in Figure 3, our results show that adding fading memory strictly improves pre-training perplexity over the baseline hybrid model at all scales. Performance is further improved when eidetic memory is added despite incurring a slightly higher training time. Our results suggest that B'MOJO exhibits a non saturating scaling law: increasing the amount of resources (parameters/FLOPs) leads to increasingly better models.

## 4.3 Zero Shot Evaluation

We catalog the performance of our pre-trained models on an assortment of short and long context zero-shot evaluation tasks. While perplexity captures the models' ability to predict language, these evaluations characterize their generalization capabilities to unseen tasks. We use the EleutherAI LLM Harness [17] to conduct all evaluations.

**Short Context Evaluation.** In Table 1 we report results on common-sense reasoning and question-answering tasks that require processing both short [18] and medium-sized contexts [2, 34]. Since these language tasks do not require long range modeling we would not expect B'MOJO to meaningfully outperform our baselines. Moreover, B'MOJO uses a smaller sliding window (512 tokens) than Mistral (1024 tokens), placing the former on an uneven footing. Despite these caveats, we find that B'MOJO still bests Mamba at the 1.4B scale and performs comparably to the Mistral model.

**Long Context Evaluation.** We next investigate the ability of B'MOJO to process long contexts using the PG-19 dataset [17] and more recall-intensive natural language tasks using the SWDE, Scrolls [2, 40] benchmarks. Our results show that B'MOJO outperforms Mamba and our hybrid baseline on PG-19 and SWDE, while B'MOJO-F outperforms Mamba on the Scrolls datasets, in line with our previous findings on the synthetic tasks in Figure 2. These long context tasks showcase—in a practical setting—B'MOJO's efficacy in recalling information from beyond the attention span.

Table 2: **Long range downstream tasks.** We compare different architectures on several zero-shot long range recall-intensive downstream tasks [2]. Our B'MOJO variants outperform Mamba since they have stronger recall capabilities.

| | Long Context (log-ppl ↓) PG-19 | SWDE | Long Context (acc ↑) Scrolls-QAsper | Scrolls-NarraQA |
|---|---|---|---|---|
| **370M** Mistral (Full-Attention) | 2.91 | 47.16 | 13.35 | 9.24 |
| Mamba (SSM) | 3.24 | 7.38 | 9.54 | 6.04 |
| Hybrid (Sliding Attention + SSM) | 3.13 | 8.55 | 7.77 | 5.35 |
| BMoJo (Fading) | 3.05 | 15.56 | 10.29 | 7.48 |
| BMoJo (Fading + Eidetic) | 3.04 | 17.91 | 9.02 | 5.92 |
| **790M** Mistral (Full-Attention) | 2.73 | 61.2 | 14.80 | 12.29 |
| Mamba (SSM) | 2.98 | 17.37 | 12.43 | 9.62 |
| Hybrid (Sliding Attention + SSM) | 3.08 | 8.37 | 7.63 | 3.62 |
| BMoJo (Fading) | 2.83 | 22.59 | 12.8 | 9.66 |
| BMoJo (Fading + Eidetic) | 2.84 | 23.40 | 11.05 | 7.64 |

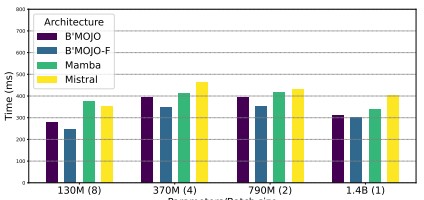

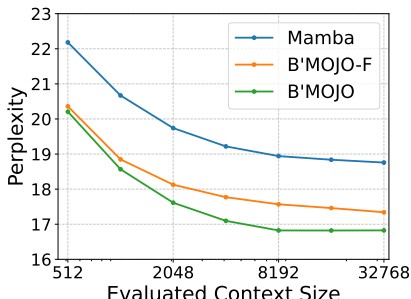

Figure 4: **Time in ms to process 2k sequences**. B'MOJO is faster than other efficient implementations of Mamba [18] and Transformers [18] at all scales.

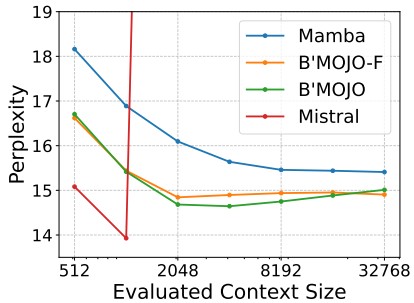

Figure 5: **Length generalization. (Left)** We pre-train B'MOJO 1.4B and Mamba 1.4B on 2k context lengths and a 1.4B Transformer baseline on 1k. **(Right)** We pre-train B'MOJO 790M and Mamba 790M on 8k context length and compare models evaluating perplexity on longer sequences up to 32K tokens on PG-19. Transformers cannot length generalize (a known failure mode), on the other hand B'MOJO preserves/improves in perplexity better than Mamba even on longer sequences.

## 4.4 Length Generalization

We evaluate the ability of B'MOJO to improve its predictions with longer contexts than ones seen during training, an attribute termed *length generalization* [10, 49, 3]. Whereas length generalization in Transformers is limited by positional encodings and memory constraints [10], for SSMs and B'MOJO, it is instead limited by the capacity of the recurrent state. In Figure 5 we report perplexity on PG-19 as the model processes contexts larger than pre-training contexts. We observe that B'MOJO 1.4B and Mamba 1.4B are capable of reducing and maintaining lower perplexity levels than Mistral at long context sizes. Curiously, we find that on models trained on longer sequences (8k), length generalization still holds and allows the model to continuously reduce perplexity as more tokens are processed, up to $4\times$ the pre-training sequence length. Additionally, the pre-training perplexity of B'MOJO models trained on longer contexts is lower than identical ones trained on the same amount of tokens but on shorter contexts, showcasing that our model can properly leverage the eidetic and fading memory to process long sequences. In Appendix B.2, we report our implementation of backpropagation through time that we use to efficiently train our models on even longer contexts (beyond 16k). Our results on smaller scales (up to 370M) show that a model trained on 16k tokens can length generalize up to contexts of length 64k.

## 5 Conclusions and Limitations

Although our experiments have been conducted up to a 1.4B scale, scaling B'MOJO further requires non-trivial engineering and compute. Therefore, despite B'MOJO's promising scaling laws, it is difficult to ascertain whether it could scale to even larger models and datasets, and do so competitively. Scaling our work to even larger models could result in positive societal benefits such as ease of access to information. However, these models could be used also to spread misinformation, so novel algorithms and research is important to improve controllability and reduce hallucination [15]. Despite our promising results on length generalization, we observed that Mamba checkpoints trained on more compute tend not to length generalize that well. Since B'MOJO leverages a Mamba-like module to implement fading memory, we cannot exclude the possibility that it will be less effective in length generalization as we scale more. However, exploring simple time normalization techniques as mitigations is a promising area for future work [31].

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

# A  Induction and Transduction

**Example A.1** (Biology).  We note that biological agents have no option but to operate inductively, due to (a) hard memory bounds, and (b) evolutionary pressure towards minimizing inference latency: When faced with a threat, a biological agent is better served by a quick suboptimal decision than by reasoning over all past experience.  AI built on silicon has no such limitations: Memory can grow unbounded and test-time computation can be distributed and improved by hardware design. Nonetheless, any practical realization involves some kind of constraint on inference time or compute resources.  Therefore, resource-constrained optimal inference hinges on how to best use the available resources against a growing memory.

**Example A.2** (CNN Classifiers, VAEs and GANs).  A trained representation can be co-opted to generate data.  For example, a CNN can be used to classify random data until one is labeled with the desired class, and the resulting sample considered as being "generated" by the CNN. Similarly, one could generate random data indirectly by feeding noise to an encoder, as done in Generative Adversarial Networks (GANs), again co-opting a representation for generating data. In a Variational Autoencoder (VAE), data is generated by perturbing the latent representation of a map trained to re-construct the dataset.

**Example A.3** (Diffusion Models).  Diffusion Models are representations, trained to re-construct the original data, but the mechanics used to reconstruct the data during training are sequential, using an artificial "time" variable, akin to a realization. This makes their use as "generative representation" natural since the reconstruction process is already a stochastic realization.[8]

**Example A.4** (The Sage and the Savant).  Picture the Library of Alexandria in 40 BCE, with the two best known experts, Sage and Savant. Sage had spent years reading the entire library and distilled its content down to maxims, proverbs, and various nuggets of wisdom. When asked a question, Sage would quickly return a pithy answer, although for the occasional unusual question, a generic answer. Savant was the librarian, with a preternatural ability to rapidly find content to assemble answers to any question. Savant did not have ready answers but, when asked, Savant would scour the library to retrieve relevant sources, speed-read through them, and assemble an answer. If asked the same question by the next customer, Savant would repeat the process anew, to the dismay of customers who saw Savant re-read the same material over and over, seemingly without understanding or learning anything. When voices spread that Savant produced more accurate answers, the enraged Sage burned down the library, putting Savant out of work. Sage regained the status of preeminent source of consultation, who could generalize wisdom from sources long gone, until two millennia later, when a new Savant was created from bits.

Sage personifies *inductive learning*, favoring thoughtful and time-consuming learning to enable quick inference and rapid answers to questions. Savant represents *transductive inference*, which requires access to memory and efficient computation that is specific and tailored to the question, at the cost of having to repeat it all over.

# B  B'MOJO implementation details

## B.1  Training details

We train all the models from scratch on a common language dataset at scales from 130M to 1.4B using instances with 8 40GB NVIDIA A100 GPUs. For our 1.4B experiments, we use 8 instances simultaneously to train our model and perform the remaining experiments on a single instance. Models are trained using AdamW on 20x the number of tokens as parameters [20]. We use a batch size of 0.5M tokens and a cosine learning rate schedule with 5% warmup and a minimum learning rate of 1e-5. Finally, we use a weight decay of 0.1 and gradient clipping of 1 (see the Appendix for additional details). B'MOJO and B'MOJO-F do not use positional encodings, and both, along with Hybrid use a sliding window of 512 tokens. Furthermore, we found that the learning dynamics of our hybrid models can be improved by having two different learning rates parameters for the hidden SSM and the sliding window. We follow Mamba's learning rates [18] and GPT3's [6] for the sliding window.

---

[8]It is curious that the reverse diffusion equation was first derived in the context of Stochastic Realization Theory by [28] to argue that a stochastic realization is a *model*, distinct from the physical system it realizes: Time is reversible in the former, but not in the latter.

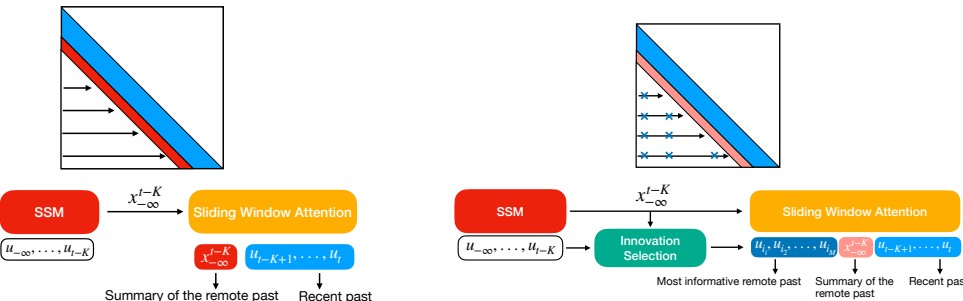

Figure 6: **B'MOJO's Memory management. (Left) Fading Memory** B'MOJO fading memory is computed by a SSM that represents long-range dependencies through a fixed-dimensional representation which is later aggregated on the current tokens along with with the most recent past. **(Right) Eidetic + Fading Memory** Fading memory is handled as in the left panel while tokens from the past are selected using an innovation test over the SSM output and appended to the current sliding window. The innovation test measures how difficult a new tokens is to predict using the state of the SSM, if a tokens is difficult to predict from the state we store it in the eidetic memory and pass it to the attention module.

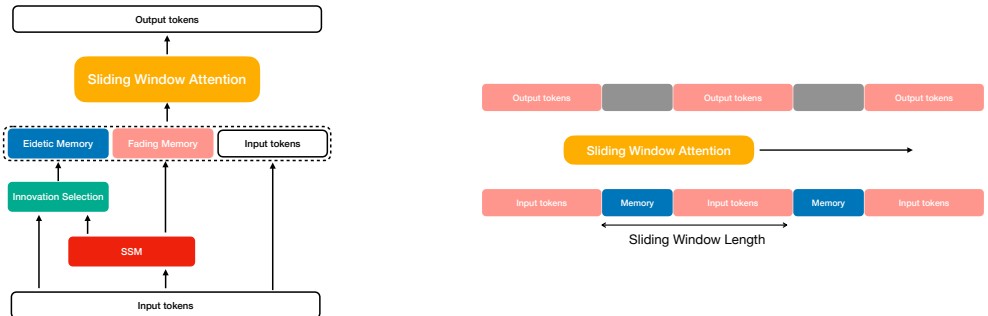

Figure 7: **(Left) B'MOJO's implementation.** We report the basic layer we use to implement B'MOJO and its memory hierachy (fading and eidetic). **(Right) Efficient interleaved implementation**. We show to efficiently implement a sliding window attention over chunks of input, fading and eidetic tokens.

## B.2 B'MOJO's efficient implementation

We efficiently implement B'MOJO's tiered memory hierarchy at scale. The modelling choices that lead to B'MOJO closely follow ideas from Stochastic Realization and have an obvious recurrent efficient implementation. On the other hand, during training one is more interested in a parallel formulation that can allow to process multiple tokens at the same time in a single sequence. In the following we describe how we use chunking to develop B'MOJO's parallel form.

**Efficient sliding window.** Computing fading memory and selecting the most unpredictable tokens to store before feeding them into an attention layer is a sequential process which is hard to parallelize. To solve this we use chunks of length $w$ and use fading and eidetic memory to summarize the whole past before the beginning of each chunk. Then, to efficiently aggregate information from the input and memory tokens we use a sliding window over the interleaved concatenation of input chunks and their respective memory tokens as shown in Figure 7. To make sure that every token is computed uniformly we pick the size of the sliding window to be $K := w + m_f + m_e$, where $m_f$ is the number of fading memory tokens, and $m_e$ is the number of eidetic tokens. Note that the number of eidetic tokens is not known a priori however, in our experiments, we fixed it to an upper-bound. The modular and interleaved approach Figure 7 enables us to leverage optimized kernels for both the SSM [18] and the sliding window attention [9] components enabling fast training and inference beyond the 1B scale. Furthermore, one can accelerate the Attention components in B'MOJO using standard

techniques like GQA, which, for example, can make B'MOJO 1.4B with 2k context length up to 7% faster than a vanilla implementation.

**Efficient Innovation Selection.** The Innovation Selection process we describe in Algorithm Line 1 requires a $\hat{y}(y_{t-1:t-k})$ whose task is to predict the next output $y_t$ of the SSM. Whenever, it is difficult to predict $y_t$ using samples from the past $y_{t-1:t-k}$ the consider the most recent input token $u_t$ highly informative and store it. While building a new parametric predictor with learnable parameters is possible, in practice, this will require to modify the training loss. In our experiments we consider a fixed predictor so no extra weights need to be learned. We fix $\hat{y}$ as a running weighted average and implement it using short 1D grouped causal convolutions.

**BPTT: Back-propagation Through Time** In this section we discuss our technique for back-propagation through time which enables training with arbitrary context length, potentially infinite. This enables us to treat the training data as batch of samples or as one large string. The trick to enable such training lies in splitting the data into multiple chunks, performing computations on each chunk, caching the statistics like the hidden states for SSMs or the tokens in the previous sliding window/eidetic memory, and then using them for the next chunk of data. One way to efficiently implement this is to write a custom CUDA kernel for this task which defines a Mamba layer which can process a non-zero initial hidden state (as the current implementations only support a zero initial hidden state). However, we would like to use the existing implementations and instead modify the inputs/layer weights such that we can load the cached hidden state before that start of every chunk. Let $x_t \in \mathbb{R}^{b \times d \times h}$ be the hidden state of an SSM layer (Mamba), where $b, d, h$ are the batch-size, model hidden dimension, state hidden dimension (channels for each state dimension) respectively, and $u_t \in \mathbb{R}^{b \times d}$ is the input to the layer. The state update equations in Mamba evolve as $x_{t+1} = A(u_t) * x_t + B(u_t) * \widehat{u_t}$, where $A(u_t) \in \mathbb{R}^{b \times d \times h}$, $B(u_t) \in \mathbb{R}^{b \times d \times h}$, $\widehat{u_t}$ is $u_t$ stacked $h\times$, and $*$ is an element-wise multiplication operation. In the traditional implementations $x_0 = 0$, however, we would like to use a non-zero initial state $x_0'$. The trick we use here is to break $x_0' \in \mathbb{R}^{b \times d \times h}$ into $h$ tokens of size $b \times d$ (which is the same as the size of $u_t$) and pass them as additional prompts to the input. In practice the dimension $h$ is usually small (like 16) and as a result we can afford to break the hidden state into tokens and pass them as inputs. Let $T$ be the size of each chunk (input), then to enable processing non-zero initial hidden states (or caching) we need to process chunks of size $T + h$ where $T \gg h$. We perform $h$ initial steps of the SSM to load the previous hidden state, before we start processing the current chunk. To dynamics for the first $h$ timesteps are governed by the following equation for $t \in [0, h]$: $x_{t+1} = x_t + B_t' * \widehat{x_{0,t}'}$, where $x_{0,t}'$ chooses elements from the $t$ column in the final dimension, $B_t'$ is one-hot with 1 in dimension $t$, and acts a shifting operation which ensures that the tokenized previous hidden state is correctly loaded. Thus after $h$ time-steps the hidden state $x_h$ is loaded with the final hidden state from the previous chunk, which can be used with gradient accumulation techniques to compute gradients on arbitrarily sized input sequences. Note that B'MOJO has additional components like the eidetic memory, however, that can be efficiently cached from the previous chunk in constant time/memory cost.

## C State Space realization Attention

Lets start from the usual expression of the attention,[9] given an input sequence $\{u_i\} \in \mathbb{R}^{n_{ch}}$, we create the keys, values and queries vectors as follows, $k_i := \{W_K u_i\} \in \mathbb{R}^n$, $q_i := \{W_Q u_i\} \in \mathbb{R}^n$, $v_i := \{W_V u_i\} \in \mathbb{R}^n$. Now the output of a causal attention layer is given by:

$$y_t = \frac{\sum_{i=1}^{t} \exp(q_t^T k_i) v_i}{\sum_{i=1}^{t} \exp(q_t^T k_i)} \tag{2}$$

### C.1 Linear attention

It is possible to approximate the exponential in the numerator using a kernel representation which makes the attention a linear operator (the ratio of two linear operators) onto the augmented feature space. The basic idea is to write $\exp(q_t^T k_i) \approx \phi(q_t)^T \phi(k_i)$, we therefore get that the linear attention can be computed as:

$$y_t \approx y_t^{lin} = \frac{\sum_{i=1}^{t} \phi(q_t)^T \phi(k_i) v_i}{\sum_{i=1}^{t} \phi(q_t)^T \phi(k_i)} \tag{3}$$

---

[9]For the sake of simplicity we do not use Multi-Head Attention, however the generalization is straightforward.

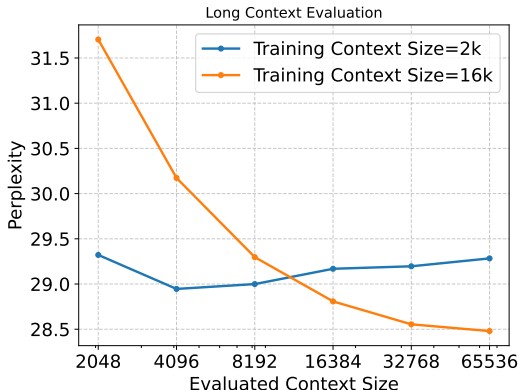

Figure 8: **B'MOJO's long context training using BPTT Appendix B.2**. We train B'MOJO with two different context sizes, 2k and 16k (BPTT) respectively and evaluate on long context task (PG-19). We show that our model trained with 2k context size is able to extrapolate for context size upto 65536 (with marginal increase in the perplexity), while model trained with 16k context size can handle long context much more effectively which can be seen by the lower perplexity values as the context size increases. Remarkably, as previously observed in [10], B'MOJO models trained on long contexts underperform models trained on shorter ones if evaluated on fewer tokens (i.e., when the inference context size is much smaller than the training context size).

Now consider the numerator, the denominator can be obtained by fixing the input $v_i = 1 \; \forall i$. It is trivial to show that we can represent the linear attention using a Finite-Impulse Response dynamical system as follows:

$$\begin{cases} x_t = \sum_{i=1}^{t} \phi(k_i) v_i \\ y_t = \phi(q_t) x_t \end{cases} \tag{4}$$

This is a linear dynamical system that evolves over time and returns the final values of the linear attention at each time instant $t$.

**Remark**: Note that we can modify the previous equations to represent the attention mechanism by simply using a non-linear read out function (the $\exp$) in place of the identity. However, this is not very insightful since this simple FIR system is simply a shifting register over time.

It is worth noticing that the state of this system is $t$, and it always increases over time, usually up to a design parameter specified by the system design which is dictated by the available compute and memory (typical values are set to 2k).

A simple way to measure the state of the system is to write its state space realization, whose dimension directly informs us on the expressivity of the system. In our case we shall assume that the attention is computed on a sliding window of size $K$. A canonical realization of the FIR described above is:

$$\begin{cases} Z_{t+1} := \begin{bmatrix} z^{t-K+1} \\ \vdots \\ z^{t+1} \end{bmatrix}_{t+1} = \begin{bmatrix} 0 & 1 & & \\ & & \ddots & \\ & & & 1 \\ 0 & 0 & \dots & 0 \end{bmatrix} \begin{bmatrix} z^{t-K} \\ \vdots \\ z^t \end{bmatrix}_t + \begin{bmatrix} 0 \\ 0 \\ \vdots \\ 1 \end{bmatrix} \phi(k_t) v_t \\ \bar{Z}_t = \begin{bmatrix} 1 & \dots & 1 \end{bmatrix} Z_t + \phi(k_t) v_t \\ \mathbf{y}_t^{lin} = \phi(q_t) \bar{Z}_t \end{cases} \tag{5}$$

## C.2 Connection with State Space Models

In this section, we describe the connection with State Space models, in particular Mamba (input dependent state space model) and characterize how it approximates a linear attention mechanism.

First we state the Mamba equations:

$$\begin{cases} \bar{u}_t = w_0 u_t + w_1 u_{t-1} + w_2 u_{t-2} + w_3 u_{t-3} \\ x_t = a(\bar{u}_t)x_{t-1} + b(\bar{u}_t)\bar{u}_t \\ y_t = c(\bar{u}_t)x_t + d\bar{u}_t \end{cases} \tag{6}$$

For the sake of simplicity we shall now study the input to state and the read out equation. It is easy to show that this is not a strictly causal realization of a dynamical system (since the state at time $t$ is updated with the input at the same time).

**Remark**: Every single channel in a Mamba block is independent from each other and unnormalized, Mamba reduces the variability across channels with a coupling in the lower dimensional projections of the gating parameter $\Delta$.

### C.2.1 Local Global factorization of the attention mechanism

A natural way to prevent the system matrices to grow ever larger is to approximate the FIR above using an AR component, which would essentially keep a running average of the keys for each token seen so far. This will allow the model to keep some higher level statistics of past data which would be used to summarize past information into a single dimension of the dynamical system.

$$y_t = \sum_{i=1}^{t} \phi(q_t)^T \phi(k_i) v_i = \phi(q_t)^T \Big( \sum_{i=0}^{K-1} \phi(k_{t-i}) v_{t-i} + \sum_{i=1}^{t-K} \phi(k_i) v_i \Big) \tag{7}$$

$$= \phi(q_t)^T \Big( \sum_{i=0}^{K-1} \phi(k_{t-i}) v_{t-i} + \bar{Z}_{t-K} \Big) \tag{8}$$

which can be written as:

$$\begin{cases} x_t = x_{t-K} + \sum_{i=0}^{K-1} \phi(k_{t-i}) v_{t-i} \\ y_t = \phi(q_t)^T x_t \end{cases} \tag{9}$$

and canonically realized as:

$$\begin{cases} Z_{t+1} := \begin{bmatrix} z^{t-K+1} \\ \vdots \\ z^{t+1} \end{bmatrix}_{t+1} = \begin{bmatrix} 0 & 1 & & \\ & & \ddots & \\ & & & 1 \\ 0 & 0 & \cdots & 1 \end{bmatrix} \begin{bmatrix} z^{t-K} \\ \vdots \\ z^t \end{bmatrix}_t + \begin{bmatrix} 0 \\ 0 \\ \vdots \\ 1 \end{bmatrix} \phi(k_t) v_t \\ \bar{Z}_t = \begin{bmatrix} 1 & \cdots & 1 \end{bmatrix} Z_t + \phi(k_t) v_t \\ y_t = \phi(q_t) \bar{Z}_t \end{cases} \tag{10}$$

### C.3 Elements of Realization Theory

When studying dynamical systems it comes particularly helpful to study canonical forms. They are particularly well suited to assess properties of dynamical systems and to realize state space models that realize a desired input-output behavior.

In particular, given a transfer function of a LTI system

$$W(z) = \frac{\beta_0 + \beta_1 z + \ldots + \beta_{n-1} z^{n-1}}{\alpha_0 + \alpha_1 z + \ldots + \alpha_{n-1} z^{n-1} + z^n} \tag{11}$$

we can write its state space canonical controllable realization as

$$\begin{cases} X_{t+1} = \begin{bmatrix} 0 & 1 & & \\ & & \ddots & \\ & & & 1 \\ -\alpha_0 & -\alpha_1 & \cdots & -\alpha_{n-1} \end{bmatrix} X_t + \begin{bmatrix} 0 \\ 0 \\ \vdots \\ 1 \end{bmatrix} u_t \\ y_t = \begin{bmatrix} \beta_0 & \cdots & \beta_{n-1} \end{bmatrix} X_t + Du_t \end{cases} \tag{12}$$

The first simple result to show is that any causal (but non-strictly) dynamical system in the following form

$$\begin{cases} x_t = Ax_{t-1} + Bu_t \\ y_t = Cx_t + Du_t \end{cases} \tag{13}$$

can be rewritten in its canonical form as

$$\begin{cases} \hat{x}_t = A\hat{x}_{t-1} + u_{t-1} \\ y_t = \hat{C}\hat{x}_t + \hat{D}u_t \end{cases} \tag{14}$$

In fact, starting from the transfer function of the first system

$$W(z) = \frac{B}{1 - Az^{-1}} = \frac{zB}{z - A} = B + \frac{AB}{z - A} \tag{15}$$

We get

$$\begin{cases} \hat{x}_{t+1} = A\hat{x}_t + u_{t-1} \\ x_t = AB\hat{x}_t + Bu_t \\ y_t = Cx_t + Du_t \end{cases} \rightarrow \begin{cases} \hat{x}_{t+1} = A\hat{x}_t + u_{t-1} \\ y_t = CAB\hat{x}_t + (CB + D)u_t \end{cases} \tag{16}$$

Note that the terms $C$ and $B$ only appear as the product $CB$, which then we can rename as $\hat{C}$.

**Remark**: Extending the previous canonical from to Time-Varying Systems is tedious but straightforward.

The controllable canonical form we introduced in Equation (12) can be easily extended to Time-Varying Systems and Input-Varying as well since it encodes the algebraic properties of the relationship between "positional" variables at time time instants (or of different inputs). Hence, it is straightforward to see that when setting all the coefficients on the last row of the state transition matrix to zero we get back the same nilpotent dynamical system that represents the attention mechanism (note that, differently from the linear attention case, the read-out function is non-linear).

On the other hand, we can use this canonical form to represent a Mamba (diagonal) model (which is non-canonical) by simply picking the coefficients of the characteristic polynomial such that its poles are the same as the diagonal entries of the Mamba block. Note however, that Mamba being non-minimal, could have some zero-pole cancellations depending on the specific values of the input matrix $B(u_t)$ and $C(u_t)$, in such cases the input-output behaviour associated with the cancellation does not appear in the output of Mamba and, so long as it is stable (which is always the case thanks to Mamba's parametrization), it is guaranteed to remain bounded and decay to zero exponentially fast.

Hence, both Mamba and the Attention mechanism can be implemented by B'MOJO for any fixed state $N$.

## D  Further empirical results

### D.1  Synthetic Tasks Beyond Associative Recall

In Table 3 we report results on synthetic tasks other than Multi-Query Associative Recall (MQAR). The table below expands the range of synthetic tasks from MQAR to four more synthetic tasks and compares performance across multiple scales.

### D.2  Scaling laws and zero-shot evaluations

In Figure 3, we report perplexity at different scales against (left) the number of parameters and (right) the wall-clock training time. B'MOJO is faster than Mamba and Mistral that use efficient CUDA kernels (Flash attention and Selective Scan), outperforms our pre-trained Mamba baseline at all scales, and is comparable with our Mistral Transformer model.

While prior works like Jamba [27] slightly outperform pure Transformer models they used full attention, and thus retain the quadratic dependence of Transformers. In contrast, our work only uses a small 512 token sliding window attention to produce a model with linear dependency on the sequence length with a constant KV cache size. Other works like Griffin [10] also use sliding windows, but manage to slightly outperform Transformer leveraging much longer sliding windows sizes than ours.

Table 3: **BMOJO demonstrates strong performance on a wide range of synthetic tasks, and both small and medium scale.** We compare the performance of various models at the 2 layer and the 130M scale on 4 different synthetic tasks, (1) Selective Copying, a task involving recall of a specific sequence of tokens with random spacing (2) Induction Heads, the recall of a specific token amongst noisy tokens (3) Noisy MQAR, an associative recall task retrieving keys in a noisy environment and (4) Fuzzy Recall, an associative recall task involving keys and values that are multiple tokens each. We find that B'MOJO models consistently outperform or match all existing baselines.

| Model | Selective Copying | | Induction Heads | | Noisy MQAR | | Fuzzy MQAR | |
| --- | --- | --- | --- | --- | --- | --- | --- | --- |
| | 2 layers | 130M | 2 layers | 130M | 2 layers | 130M | 2 layers | 130M |
| **Full context** | | | | | | | | |
| GPT2 | 93.56 | 97.28 | 100 | 1 | 99.92 | 99.97 | 49.17 | 98.79 |
| Mistral | 94.67 | | 100 | | 99.99 | | 98.23 | |
| Pythia 160m | | 99.9 | 100 | 1 | | 1 | | 98.99 |
| **SSM** | | | | | | | | |
| Mamba | 94.42 | 99.74 | 100 | 7.6 | 99.99 | 1 | 88.04 | 60.04 |
| **Reduced context (smaller window)** | | | | | | | | |
| Hybrid | 93.82 | 97.69 | 7.64 | 1 | 99.99 | 99.98 | 90.47 | 98.56 |
| B'MOJO-F | 93.92 | 98.58 | 100 | 1 | 99.97 | 99.99 | 90.62 | 98.35 |
| B'MOJO | 94.04 | 99.85 | 99.94 | 1 | 99.99 | 99.99 | 90.38 | 96.87 |

# E   Strip MAMBA

In this section we derive the form of MAMBA reported above from the original source, combining the published paper and software implementation provided by the authors [18].

Every Mamba layer contains a State Space Model which maps sequences $(B, L, d_{in})$ to $(B, L, d_{in})$. Call the input of the state space models in Mamba is $(B, L, d_{in})$ is $\{u_i\}_{i=0}^L$.

The input sequence is used to generate the discretization time-step $\Delta$, the input matrix $B$ and the output matrix $C$, using the following projection matrices: $P_{\Delta_{down}} \in \mathbb{R}^{\Delta_{down} \times d_{in}}$, $P_{\Delta_{up}} \in \mathbb{R}^{d_{in} \times \Delta_{down}}$ and $P_B \in \mathbb{R}^{N \times d_{in}}$ and $P_C \in \mathbb{R}^{N \times d_{in}}$.

Overall, we have

$$\begin{cases} B(u_t) = P_B u_t \in \mathbb{R}^{d_{in}} \\ C(u_t) = P_C u_t \in \mathbb{R}^{d_{in}} \\ \Delta(u_t) = \text{softplus}(P_{\Delta_{up}} P_{\Delta_{down}} u_t) \in \mathbb{R}^{d_{in}} \end{cases} \tag{17}$$

Now, given a state representation for all $d_{in}$ dimensions $x_t \in \mathbb{R}^{d_{in} \times N}$ and the state update matrix $A \in \mathbb{R}^{d_{in} \times N}$, We can now write the mamba update rule as:

$$\Delta(u_t) = \text{softplus}(P_{\Delta_{up}} P_{\Delta_{down}} u_t) \in \mathbb{R}^{d_{in}}$$
$$x_{t+1} = \exp(\text{Diag}(\Delta(u_t))A) * x_t + \text{Diag}(\Delta(u_t)) u_t u_t^T P_B^T$$
$$y_t = x_t P_C^T u_t + D$$

where $*$ denotes the element wise product. The elementwise product makes clear that Mamba's dynamics are diagonal. Revisiting the description of Mamba's diagonal dynamics in the main paper,

$$A(u_t) = A_{\text{Mamba}}(u_t) = \begin{bmatrix} a_1(u_t) & & \\ & \ddots & \\ & & a_N(u_t) \end{bmatrix}, \quad b(u_t) = b_{\text{Mamba}}(u_t) = \begin{bmatrix} b_1(u_t) \\ \vdots \\ b_N(u_t) \end{bmatrix}$$

it is natural to see that the values of $a_i(u_t)$ are elements of the matrix $\exp(\text{Diag}(\Delta(u_t))A) = \exp(\text{Diag}(\text{softplus}(P_{\Delta_{up}} P_{\Delta_{down}} u_t))A)$, and $b_i(u_t)$ are elements of $\text{Diag}(\Delta(u_t)) u_t u_t^T P_B^T$. Our non-linear readout function $\rho$ is simply bilinear in $x_t$ and $u_t$.

### E.1 Bilinear Mamba

With some approximations, we can further show that Mamba can be written as a bilinear system on an augmented input space. Using the first order Taylor Series, we have

$$
\begin{aligned}
x_{t+1} &= e^{\mathrm{Diag}(\Delta_t)A} * x_t + \mathrm{Diag}(\Delta_t)u_t u_t^T P_B \\
&= (\mathrm{Diag}(\Delta_t)A + \mathbf{1}\mathbf{1}^T) * x_t + \mathrm{Diag}(\Delta_t)u_t u_t^T P_B \\
&= x_t + \mathrm{Diag}(\Delta_t)(A * x_t + u_t u_t^T P_B) \\
&= x_t + \mathrm{Diag}(\log(1 + \exp(P_\Delta u_t)))(A * x_t + u_t u_t^T P_B) \\
&= x_t + \mathrm{Diag}([P_\Delta u_t]_+)(A * x_t + u_t u_t^T P_B)
\end{aligned}
$$

where the second line follows from the application of the first order Taylor Series, the third from rearranging terms while applying the definition of $\Delta_t$, and the forth line follows from replacing notation for the soft hinge loss gating with $[\cdot]_+$. Now, we carry out some algebra as follows:

$$
\begin{aligned}
x_{t+1} &= x_t + \mathrm{Diag}([u_t]_+)(A * x_t + P_\Delta^\dagger u_t u_t^T P_\Delta^{T\dagger} P_B) \\
&= x_t + \mathrm{Diag}([u_t]_+)A * x_t + \mathrm{Diag}([u_t]_+)P_\Delta^\dagger u_t u_t^T P_B \\
&= x_t + \mathrm{Diag}([u_t]_+)A * x_t + (u_t^T \otimes \mathrm{Diag}([u_t]_+))\mathrm{vec}(P_\Delta^\dagger)\mathrm{vec}(P_B)^T(I \otimes u_t^T)^T
\end{aligned}
$$

where vec vectorizes a matrix into a column vector, and $\otimes$ is the Kronecker Product between two matrices. Now we can look at the vectorized evolution of $h$, denoting $\mathrm{vec}(x_t)$ as $\tilde{x}_t \in \mathbb{R}^{dN}$:

$$
\begin{aligned}
\tilde{x}_{t+1} &= \tilde{x}_t + \mathrm{vec}(\mathrm{Diag}([u_t]_+)A) * \tilde{x}_t + \mathrm{vec}((u_t^T \otimes \mathrm{Diag}([u_t]_+))\mathrm{vec}(P_\Delta^\dagger)\mathrm{vec}(P_B)^T(I \otimes u_t^T)^T) \\
&= \tilde{x}_t + \underbrace{(A^T \odot I)}_{Parameters}\underbrace{[u_t]_+}_{Features} * \tilde{x}_t + \underbrace{((I \otimes u_t^T) \otimes (u_t^T \otimes \mathrm{Diag}([u_t]_+)))}_{Features}\underbrace{(\mathrm{vec}(P_B) \otimes \mathrm{vec}(P_\Delta^\dagger))}_{Parameters},
\end{aligned}
$$

where $\odot$ is the Khatri-Rao product. The purpose of this highly messy derivation is simply to demonstrate that there exists a feature map of $u_t$ that we will call $z_t$, and sparse parameter sets $\tilde{A}$, $\tilde{P}_B, \tilde{P}_C$ such that

$$
\begin{cases}
\tilde{x}_{t+1} = \tilde{A}z_t * \tilde{x}_t + z_t \tilde{P}_B \\
y_t = \tilde{x}_t \tilde{P}_C^T z_t + D,
\end{cases}
\tag{18}
$$

and the system can be described as a discrete bilinear system.

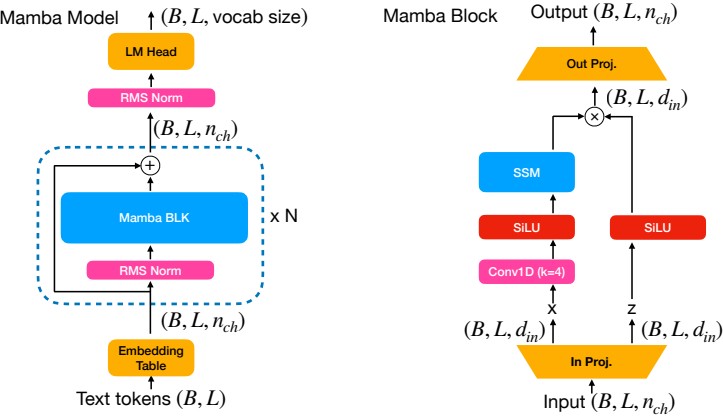

Figure 9: An illustration of the Mamba architecture and main block.

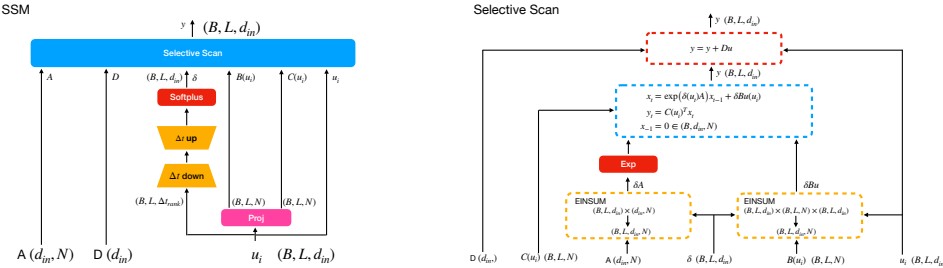

Figure 10: An illustration of the Mamba SSM block and Selective Scan operation.

