# OpenReview forum: "B'MOJO: Hybrid State Space Realizations of Foundation Models with Eidetic and Fading Memory"
_NeurIPS.cc/2024/Conference — NeurIPS 2024 poster_

### Official Review · Reviewer_MJ98 · 2024-06-23

**Soundness:** 2
**Presentation:** 2
**Contribution:** 2
**Rating:** 5
**Confidence:** 4

**Summary:**

Introduces a class of models called B'MOJO which combines sliding window attention with SSMs. The main version of B'MOJO leverages an error function to decide what gets added to the long term sliding KV cache, inspired by ideas from Stochastic Realization Theory. Experiments are performed on multi-query associative recall (MQAR) and well as language modeling.

**Strengths:**

- Considers the important and relevant problem of balancing memory and efficiency in modern sequence models
- Provides a nice, high level overview of stochastic realization theory and its connections with modern sequence models
- The idea of using an error function inspired by the idea of an innovation process to determine a storage/eviction policy in sliding window attention appears to be novel

**Weaknesses:**

- There are several related lines of work that are not discussed well enough and/or should be cited:
    - Block state transformers: https://arxiv.org/abs/2306.09539 and SeqBOAT: https://arxiv.org/abs/2306.11197, considers using an SSM to contextualize information that is input to a block-windowed/sliding window attention mechanism, quite similar to the setup in this paper
    - GSS (https://arxiv.org/abs/2206.13947), H3 (cited in paper but not in the context of hybrids) both propose combining attention with SSMs to try and balance eidetic and fading memory, while hybrid attention methods such as Longformer (https://arxiv.org/abs/2004.05150) are also relevant
    - Works on adaptive kv caches and optimal eviction policies are also relevant, a few but incomplete list of examples: https://arxiv.org/abs/2310.01801,https://arxiv.org/abs/2401.06104, https://arxiv.org/abs/2402.18096, https://arxiv.org/abs/2402.06262
  - There is a brief discussion of input dependency in recent SSMs in Line 135, but https://arxiv.org/abs/2209.12951 should also be cited here as it was the first of this recent wave of linear RNN/SSM papers to propose efficient input-varying systems. https://arxiv.org/abs/2311.04823 is also relevant here. More generally, it could be helpful to point the reader to the SSM papers that led to the more recent variants mentioned in line 135, e.g. S4 (https://arxiv.org/abs/2111.00396), S5 (https://arxiv.org/abs/2208.04933), LRU (https://arxiv.org/abs/2303.06349) and their variants to better position this work in its proper context.

- While the paper does a nice job framing the modern sequence models within the ideas of Stochastic Realization Theory and presenting its history, in my opinion, it fails to really justify what insights are being gained from this framework. It discusses fading memory (as in SSMs) and eidetic memory (as in attention) and then proposes a way to combine these, but the result is not that different from the hybrids mentioned above, which were motivated by the same thing.
  - I think the opportunity for this was in the Innovation Selection process, however the actual version discussed in lines 219-223 does not seem to be described very well. See Questions below.
  - I have other questions below which may help me better appreciate the value of this framework.
- The experimental results seem to generally be weak
  - MQAR:
     - Too many details are missing to judge the usefulness of this experiment. E.g. What task sequence length vs sliding window lengths were used?
    -  It does seem the B MOJO models provide an edge compared to the other efficient baselines, but I am skeptical since the BMOJO-F appears to do almost as well as B-MOJO. Why is this? information about sequence lengths and sliding window lengths would help to better understand this.  What happens as the task is made even more difficult with more key value pairs and longer sequences (compared to the sliding window length)?
  - Language task:
     - What dataset was used? I do not see this information anywhere.
     - It would have been nice to see a hybrid model with a few full attention layers (not just sliding attention), to better asses the potential performance vs compute advantages that are being claimed.
    - The Downstream tables are hard to read since bolds and underlines are not used to denote top scores, but in general B'MOJO's performance underperforms Transformers and also seems to not significantly outperform Mamba.
    - It can be difficult to asses how much actual exact recall vs parametric knowledge is used to perform the  long context tasks considered. Perhaps using some previously proposed recall stress tests for pretrained models such as passkey/phonebook retrieval (e.g. as in https://arxiv.org/abs/2402.19427, https://arxiv.org/abs/2402.01032 ) or some of the more difficult needle in a haystask tasks proposed in https://arxiv.org/abs/2404.06654 would help to better assess B'MOJO's long context recall ability.

**Questions:**

Many of my major questions are listed in the weaknesses. Here are a few others.

Major:
- Could you please make the difference between B'MOJO-F and B'MOJO more explicit? This is not explained well.  I think B'MOJO-F is a simple combination of SSM and Sliding sindow attention? What is the difference between this and the Hybrid baseline in the experiments?
- Could you please explicitly explain what is the actual innovation selection mechanism used and what is its motivation? The paper has a nice buildup of ideas from stochastic realization theory, but then in lines 219-223 just says a short convolution is used. How long of a convolution? What error function? What is the motivation for why this could work? It would improve the paper it the connection with the motivation from  stochastic realization theory was better developed.
- It is not explained well why B'MOJO is so much faster than Mamba or the Transformer baseline, since I think it uses the Mamba and Flash attention kernels. Is this related to the sequence lengths considered or the chunking? Why would it be faster than Mamba?

Other:
- I wouldn't refer to the Transfomer baseline as "Mistral" in the figures or Tables since to many readers this will sound like the models released by the company, as opposed to an architecture that is similar to their implementation that you trained.

**Limitations:**

Adequately discussed.

---

> ### Author Rebuttal · Authors · 2024-08-07
>
> Thank you for your feedback and your insightful comments. We've addressed the main concerns shared with other reviewers in the global comment, here we will address specific concerns.
>
> **Related work References.** We are grateful for the references and associated insights, we have revised our manuscript to incorporate and discuss these works. We acknowledge that there are many relevant related works, our goal is to not propose yet another one, but rather a general framework that encompasses them using Stochastic Realization Theory.  Indeed, the referenced hybrid models are special cases of B’MOJO, for example Block State Transformers can be obtained by removing B’MOJO’s the eidetic memory. Similarly, GSS can be obtained by removing the eidetic memory and some of the sliding window attention layers. We have incorporated more discussions in our related work section.
>
> **Unique Insights Gained From Our Framework.** 1) Long term Eidetic Memory is unique to our work and provides crucial performance gains over models without it (see Figure 3 for perplexity gains at scale with eidetic memory, and Table A1 with results on NVIDIA’s RULER benchmark). 2) Our use of interleaved memory chunking, efficient innovation selection and input dependent SSMs (like Mamba) as opposed to slow long convolutions (like the ones used in Block-State Transformers) are specific innovations to ensure our method runs efficiently on modern hardware (see Section 3.3 and Appendix B2). These are important differentiators which we hope could inspire further developments in hybrid SSM architecture design.
>
> **Difference between B'MOJO/B’MOJO-F and Hybrid baseline.** Succinctly, a vanilla hybrid layer can be written as “output = Attention(SSM(input_tokens))”, while our B’MOJO-F layers can be expressed as “output = Attention(Cat(Inputs, SSM(past inputs)))”. In particular, B’MOJO-F is not a simple stacking of SSM (Mamba) and Attention as in our baseline hybrid model and some recent hybrid models in the literature (like Jamba/Griffin). While vanilla hybrid models are limited to interleaved stacking of SSM and Attention layers, B’MOJO-F allows the attention module to attend to both the input tokens (to the SSM) and the output representations of the SSM (the fading memory). A mechanism that allow the Attention layer to merge information from "memory tokens" to the layers' input tokens.
> Differently from B’MOJO, B’MOJO-F does not use the innovation selection mechanism (no long term eidetic memory). Please see Figure 6 in the appendix for more details.
>
> **Motivation of the Innovation Selection Mechanism.** The "state" of a stochastic realization is a sufficient statistic for (next token) prediction, meaning that the state is as good as the entire past for the purpose of minimizing the prediction error (i.e. making the prediction error unpredictable). Thus, the predictability of the prediction error, which is what the innovation test measures, is the natural test to know if the state has captured the past well enough. In practice, whenever the innovation test measures high residuals we know that the state must be refreshed with new information which in turn will help the predictor in reducing future prediction residuals.
>
> **Innovation Selection's efficient implementation.** To have an efficient implementation we picked the squared loss and coupled it with a linear predictor which takes as input the most recent past states. However, having a learnable linear predictor would increase the number of parameters of the model and it would require to modify the training loss. To avoid this, we restrict the predictor to a moving average which we implement with short 1D grouped convolutions. To preserve efficiency, we leverage the same 1D convolution implementation used in the Mamba layers which uses a kernel size of length 4.
>
> **Experimental Results seem to be generally weak**.
> When performing apples-to-apples comparisons, our largest B’MOJO model (1.4B) outperforms Mamba 1.4B in pre-training perplexity by 10% and by 1% on zero shot downstream evaluations  (see Table 1, Figures 3, 5). Furthermore, as suggested by the reviewers we assessed the long context recall capabilities of B'MOJO on a more difficult needle-in-a-haystack benchmark (see the global comment). In these recall intensive tasks we found that B'MOJO significantly outperforms Mamba by 4% accuracy @2k context sizes and by 11% @4k. As well as outperforming our Transformer model on longer contexts sizes.
>
> **Synthetic MQAR experiments.**
> We use a sequence length of 256 and an attention window length of 32 so that key-value pairs stay outside the sliding window context. We tried to strike a balance between detail and readability. We plan to make our work reproducible not just by listing numbers, but by releasing our source code upon completion of the review process. As we show in Figure 2, Panel 4, adding eidetic memory tokens (the difference between B’MOJO and B’MOJO-F) steadily increases performance.
>
>
> **Why is B'MOJO faster than Mamba and the Transformer baseline?**
>  B’MOJO is faster than the Transformer baseline because it replaces MLP layers with Mamba layers. In Table A3, we report the profiling results for different sequence lengths and show that MLP layers are slower than Mamba layers as the sequence length increases.
> Furthermore, B’MOJO is faster than the Mamba baseline since it replaces half of the Mamba layers with sliding window attention layers of length <1k. In Table A3 we report the profiling results for different sequence lengths (a similar observation can be found in Figure 8 of the original Mamba paper).
>
> ### Table A3: Profiling time (in ms) of a forward call of basic blocks, measured on A100 40Gb
>
> | Context Length | 1024 | 2048 | 4096 | 8192 | 16384 |
> |----------------|------|------|------|------|-------|
> | Blocks | | | | | |
> | Mamba | 1.1 | 2.0 | 3.2 | 6.0 | 11.9 |
> | Full Attention | 1.0 | 1.9 | 5.8 | 18.8 | 94.6 |
> | MLP | 1.2 | 2.2 | 3.9 | 7.4 | 34.6 |

---

> > ### Comment · Reviewer_MJ98 · 2024-08-09
> >
> > Thank you for your comments and clarifications.
> >
> > - On providing the sliding window length and the comment _"We tried to strike a balance between detail and readability."_: Please make the sliding window length clear throughout the paper (e.g. in figure captions, experiment descriptions etc).  Understanding the sliding window length is crucial to assess the performance of the method on the recall intensive tasks, since recall tasks that fall within the sliding window length should be solved easily, so being able to solve tasks that extend beyond the sliding window length is what is interesting.
> >
> > - Regarding the note above and the new Ruler experiments: Thank you for including these. Can you please clarify again the sliding window length used for each of these experiments? Is it also possible to include the results for the trained hybrid (sliding window + attn) method? Could you also include a random guessing baseline for each task and average?
> >
> > - It is worth noting that for the 2048 context length on the new Ruler tasks, there is generally a severe dropoff in performance from the full attention method to the SSM and sliding window hybrids.
> >   - This dropoff is concerning, because often in practice, methods will not be deployed in zero shot extrapolation settings and the broader concern is high quality within the context length it was trained. This limitation is of interest to the community and should be thoroughly discussed to strengthen the paper, not ignored.
> >   - In addition, it is not entirely clear how significant the performance boost the proposed methods have over the Mamba method? Is a 1-3% boost for the 2048 context or the 2-11% boost for the 4096 sequence significant? Especially in the context of the ~28% boost the 2048 context Transformer has over the other 2048 context methods. What is the random guessing baseline?

---

> > > ### Author Response · Authors · 2024-08-10
> > >
> > > Thank you for your comments, we hope the following helps adding more clarity.
> > >
> > > **Sliding Window Length**. Our RULER experiments used 1.4B pre-trained models. The sliding window sizes are: Transformer Baseline — 2048 tokens; B’MOJO, B’MOJO-F and the hybrid baseline — 512 tokens. B’MOJO’s modules never see a sliding window longer than the Transformer’s context length.
> > >
> > >
> > > **Hybrid baseline**. We have added results in Table B1 in this comment (to complement Table A1 which we copy here). Overall, we find the hybrid model (w/512 token sliding window attention) improves over mamba by a relative 1% @2k and 3% @4k. However, it is slightly weaker than our BMOJO-F (512 token window) and significantly weaker than our full B’MOJO model (512 token window), with a relative performance gap of 7% @2k and 55% @4k.
> > >
> > > **Random baseline**. Each task in the RULER benchmark requires generating some specific subset of tokens mentioned in the context, e.g. a 5 digit number. A typical Needle-in-a-Haystack (NIAH) example follows this template: "Some special magic numbers are hidden within the following text. Make sure to memorize it. I will quiz you about the numbers afterwards. \n{context}\n What are all the special magic numbers mentioned in the provided text? The special magic numbers mentioned in the provided text are" (please see Table 2 in the RULER paper). A random baseline (in this case) has to correctly guess 5 numbers out of the vocabulary size, the probability of a correct guess is 1/(10)^5. Harder cases include multiple words, or uuids, which have even lower success probabilities — in practice we measure 0%.
> > >
> > > **Concerning Drop w.r.t Transformers**. The drop in performance from using full attention on the 2k context tokens is not concerning, but expected when using a sliding window approach that only leverages 512 tokens. Indeed, as you note above “being able to solve tasks that extend beyond the sliding window length is what is interesting.”; we agree, the Transformer baseline is the paragon in the 2k setting. To further show this, we also evaluate our models on smaller context sizes 512 and 1024, see results in the table below. At size 512, the gap with full attention is indeed null and the gap increases only slightly at size 1024. However, longer contexts set B’MOJO apart from a Transformer model: the latter’s recall performance goes to zero if tested on a context length longer than its attention span, while our models still can recall information from contexts that are up 8x longer than the attention span.
> > >
> > > **Extrapolation not often deployed in practice**. Although it is true that often in academic benchmarks, the information supporting the query fits in context, this is not true in many business applications, where the relevant context can be thousands to millions of documents, lines of code, metrics, tables, datasets, and other data that would most definitely not fit in 2048 tokens. With B’MOJO, we are developing a class of models that can cover this long tail of tasks, since the Transformer does not. If in a particular application, 2048 tokens capture the majority of use cases, we would recommend that B’MOJO’s sliding window be set to that value. This way, a practitioner attains the best of both worlds.
> > >
> > > **Performance boost**. See [Random baseline] above, and [Concerning Drop w.r.t Transformers]. For Mamba, perhaps looking at relative percentage performance is more revealing. Our B’MOJO model improves over Mamba by 8.5% @2k and 140% @4k relative performance and decreases over transformer by 55% @2k and achieves ~20% accuracy on NIAH at 4k where the transformer model cannot solve the task.
> > >
> > > ### Table B1: Long context evaluation with RULER (needle in a haystack)
> > > | Context Length | Model | S-NIHA | MK-NIAH | MV-NIAH | MQ-NIAH | Average |
> > > |----------------|--------------|--------|---------|---------|---------|----------|
> > > | 512| Transformer | 100 | 100 | 100 | 100 | 100 |
> > > | | Mamba | 100 | 67 | 78 | 53 | 75 |
> > > | | Hybrid | 100 | 100 | 100 | 100 | 100 |
> > > | | BMOJO-F | 100 | 100 | 100 | 100 | 100 |
> > > | | BMOJO | 100 | 100 | 100 | 100 | 100 |
> > > | 1024| Transformer | 100 | 97 | 63 | 100 | 90 |
> > > | | Mamba | 100 | 44 | 34 | 48 | 57
> > > | | Hybrid| 100 | 53 | 42 | 89 | 71 |
> > > | | BMOJO-F | 100 | 59 | 48 | 98 | 76 |
> > > | | BMOJO | 100 | 81 | 59 | 100 | 85 |
> > > | 2048 | Transformer | 100 | 95 | 62 | 61 | 79 |
> > > | | Mamba | 100 | 32 | 29 | 28 | 47 |
> > > | | Hybrid| 90 | 35 | 34 | 31 | 47.5 |
> > > | | BMOJO-F | 90 | 36 | 35 | 31 | 48 |
> > > | | BMOJO | 90 | 45 | 37 | 33 | 51 |
> > > | 4096 | Transformer | 0 | 0 | 0 | 0 | 0 |
> > > | | Mamba | 9 | 12 | 5 | 7 | 8 |
> > > | | Hybrid | 9 | 13 | 5 | 8 | 8.75 |
> > > | | BMOJO-F | 10 | 16 | 5 | 8 | 10 |
> > > | | BMOJO | 22 | 21 | 17 | 17 | 19 |

---

> > > > ### Comment · Reviewer_MJ98 · 2024-08-10
> > > >
> > > > Thank you for the clarifications and adding they hybrid baseline to the tables.
> > > >
> > > > I personally think the importance and benefits of the extrapolation performance is overstated. To be clear on my previous comment “being able to solve tasks that extend beyond the sliding window length is what is interesting.”, my hypothesis based on other results in the literature mentioned above is that if you pretrained the Transformer on a 4096 context and pretrained the sliding window hybrid and BMOJO methods on 4096 context with 512 sliding window, the gap between the Transformer performance and the sliding window methods would be even greater than the gap between the Transformer and sliding window methods when trained on 2048 context with 512 sliding window. Similarly for training on 32K context (while maintaining a much shorter sliding window). This issue of a performance gap when in-distribution seems more practically important right now then counting on slightly improved (yet still poor performance) extrapolation abilities, since in many cases the practitioner will be better off training the Transformer on the longer context to ensure stronger performance. But this is just an opinion and hypothesis, and to be clear I am not asking the authors to rerun this experiment on longer context due to the resources required.
> > > >
> > > > I do think the rebuttal addresses most of my main concerns and I am increasing my score.

---

> > > > > ### Author Response · Authors · 2024-08-14
> > > > >
> > > > > Thank you for your extremely thoughtful feedback and remarks. We will continue to think hard about how we can refine our claims about long context inference.

---

### Official Review · Reviewer_PLgm · 2024-07-12

**Soundness:** 3
**Presentation:** 2
**Contribution:** 4
**Rating:** 6
**Confidence:** 2

**Summary:**

I appreciate the clarification about the method being specified for a single block as well as the new longer context results.  I have decided to increase my score.

---

This seems like a very interesting paper which proposes a new recurrent architecture that seeks to combine advantages of transformers and other recurrent architectures such as Mamba.  The results in this paper are quite nice, and seem to systematically outperform Mamba.  In particular, the results on OOD length generalization are great.  There's also an innovative idea of selecting the highest error tokens to add to the set of tokens to be attended to.  This is where I have some concerns with the paper.  I found it difficult to follow the distinction between the individual BMojo blocks and the stack of BMojo blocks.  I think this could easily be addressed in the algorithm block, or even better, by adding some pseudocode to make the computation easier to follow.  It's possible that it's my own fault for not understanding this, but if it could be clarified well, I'm open to raising my score.

notes from reading the paper:
  -New recurrent architecture called BMojo.
  -Adds eidetic and fading memory, aiming to outperform Mamba.
  -Substantially improved OOD length generalization.
  -Transductive inference for sample-specific inference.
  -"Unpredictable" tokens are added to a sliding window that is attended over.
  -Mamba has only fading memory.

**Strengths:**

-The introduction and exposition are both well written.
  -The improved OOD generalization results are impressive.
  -The idea of adding the most surprising tokens to be attended seems like an interesting idea.

**Weaknesses:**

-While reading this paper, I got confused about the structure of the block, and the overall architecture.  This could be my fault, but it's possible that other readers will also get confused.

**Questions:**

-In the B'Mojo algorithm block (algorithm 1), I read this as referring to a single BMojo layer, which is then stacked multiple times to yield the final architecture.  Is this correct?  If so, it's a bit odd that every layer in your architecture (even the first layer) is predicting the final output tokens (y_t)?  If so, this strikes me as fairly unusual?  I think it would help to have something like the block in Figure 9 to clarify the architecture better.

  -Could you say how the spaces $x$, $y$ are defined in a more formal sense (e.g. something $x \in \mathcal{R}^d$?  I think this wouldn't take much space and it would benefit readability.

**Limitations:**

The limitations seem to be fairly discussed.

---

> ### Author Rebuttal · Authors · 2024-08-07
>
> Thank you for your feedback and your suggestions. We've addressed the main concerns shared with other reviewers in the global comment, here we will address specific concerns.
>
> **Confused about the structure of the block, and the overall architecture.** We have made revisions to improve the clarity of our work. Please see Figure 6 and 7 in the appendix for more details.
>
> **B'Mojo algorithm block (algorithm 1) refers to a single BMojo layer.** Yes.
>
> **Odd that every layer in your architecture (even the first layer) is predicting the final output tokens.** Each block is only predicting representations that are passed to the subsequent block, and not predicting the final tokens. Only the final block predicts the final tokens.
>
> **How are the spaces, x, y defined in a more formal sense.** The spaces x and y are indeed \in \mathbb{R}^d, where d is the embedding dimension of the Mamba block.

---

### Official Review · Reviewer_Wgie · 2024-07-12

**Soundness:** 4
**Presentation:** 3
**Contribution:** 3
**Rating:** 7
**Confidence:** 4

**Summary:**

The paper presents B'MOJO, a novel building block combining the strengths of Transformers and modern SSMs. The main motivation of the paper is to develop a system capable of combining an eidetic memory, responsible for performing transductive inference via in-context learning, and a fading memory capable of storing information from the past into a finite-size (hence lossy) state. To do so, the paper shows that Transformers and Mamba can be described as different parametrizations of simple dynamical systems, the former requiring to increase its state as its span gets larger, the latter having a fixed state and hence implementing a lossy compression of the past preventing exact recall. B'MOJO results from a natural combination of the state updates of the two aforementioned models, bypassing the limitations of both. The final model ultimately consists of a sliding window attention operating on three different memory sources: the recent past (most recent input tokens), a lossy compression of the distant past (computed via an SSM update) and the most informative tokens from the distant past. The latter memory source is obtained via "Innovative Selection": the tokens that are difficult to predict given the past are stored in a - possibly unbounded up to hardware limitations - external memory. The model is compared to Transformers, Mamba and hybrid models on synthetic tasks for associative recall, language modelling scaling laws, zero-shot evaluation on short and long range tasks and length generalisation. The proposed model performs favourably compared to Mamba and comparably with Mistral on most tasks.

**Strengths:**

* The paper is well-written and generally pleasant to read.
* The idea of combining eidetic and fading memory, while not new, is very interesting and B'MOJO represents an original and elegant way to do so.
* B'MOJO is shown to integrate the benefits of transformers and modern state-space models. The model compares favourably with Mamba on most tasks. While inferior compared to Mistral7B on zero-shot tasks, B'MOJO inherits its length generalization performance from SSMs, outperforming Mistral7B in this latter case. The model is shown to be marginally better than baselines in terms of training time.

**Weaknesses:**

* Generally Mistral 7B seems to perform comparably or even better than the proposed model, with the exception of the length generalization task. In particular, Mistral is often significantly better on the zero-shot evaluation tasks, also on those involving long contexts. Do the authors have an explanation for this?
* It is not clear how the information from the various sources of memory are aggregated in the sliding-window attention mechanism. In particular, given that the size of the eidetic memory $M$ can in principle be unbounded (up to hardware limitations), it is not clear to me how such a large memory can be efficiently processed by attention. By looking at the Appendix, it appears that the number of tokens taken from $M$ is bounded. How are tokens selected from this memory? Is there a criterion to select a specific subset of them?
* Some parts of the writeup could be made clearer: for example, in the "Transformer" part in section 3.1, the dimension $V$ is not introduced before in the text. Generally it would be helpful to clearly state the dimension of each matrix/vector for the sake of clarity.

**Questions:**

See weaknesses section.

**Limitations:**

The authors discuss the limitation of their work in Section 5.

---

> ### Author Rebuttal · Authors · 2024-08-07
>
> Thank you for your feedback and your suggestions. We've addressed the main concerns shared with other reviewers in the global comment, here we will address specific concerns.
>
> **Mistral 7B seems to perform comparably or even better.** As we clarify in general comments, in a true apples-to-apples comparison in short contexts a Transformer represents a paragon, not a baseline. The primary strength of the B’MOJO model relative to Transformers is in the long context regime. However, performance in long context tasks is sensitive to scale, especially for hybrid state space models like B’MOJO, and in Table 2 we provided experiments only at a scale that is reproducible in academic environments. However, in order to address the reservations raised in this review, we also ran larger-scale experiments up to 1.4B scale (Table A1) featuring NVIDIA’s RULER benchmark using our larger models: B’MOJO-F 1.4B (with only fading memory) and B’MOJO 1.4B (with fading + eidetic memory), where we show that both B’MOJO and B’MOJO-F outperforms Transformer models on long context tasks as expected.
>
> **Eidetic memory can in principle be unbounded, is there a criterion to select a specific subset?** In the proposed implementation we bound the number of tokens to a maximum size (to keep the cost of attention manageable). However, the token span can be arbitrarily large. We are yet to explore a selection mechanism (e.g. Landmark attention).
>
> **Clearly state the dimension of each matrix/vector.** We apologize for the confusion and have revised our draft to include dimensions of each matrix and vector.

---

> > ### Comment · Reviewer_Wgie · 2024-08-14
> >
> > I would like to thank the authors for their response to my concerns and for the additional experiments.
> >
> > The rebuttal addressed most of my concerns and I decided to increase my score accordingly.

---

### Official Review · Reviewer_majr · 2024-07-15

**Soundness:** 2
**Presentation:** 3
**Contribution:** 3
**Rating:** 3
**Confidence:** 4

**Summary:**

This work presents a new module by combining eidetic memory, fading memory, and long-term eidetic memory (through an innovative selection operation). The proposed new module has strong sequence modeling capacity and high inference efficiency. The proposed new architecture achieves perplexity comparable to Transformers and SSMs with promising long-sequence ability.

**Strengths:**

* This work provides analysis for both attention and Mamba and proposes "B'MOJO-F" by combining sliding window attention and Mamba.
* In order to further improve the memorization/recall ability, the Innovation Selection is proposed to compensate for the lossy memory in fading memory and add important token information to the long-term eidetic memory. The Long-term eidetic memory can increase as the input sequence length increases but could be way more efficient than standard attention.
* The resulting architecture shows better memory efficiency on Associative Recall tasks than Mamba, although it still underperforms Transformers.

**Weaknesses:**

* Lack of sufficient ablation study. This work claims that it leverages 4 kinds of memory: short-term eidetic memory 'in-context,' permanent structural memory 'in-weights,' fading memory 'in-state,' and long-term eidetic memory 'in-storage'. Specifically, how does the short-term eidetic memory impact the capacity for recall and language modeling? Additionally, in tables 1 & 2, for "BMoJo (Fading + Eidetic)", which eidetic memory is this referring to? It seems that adding this eidetic memory does not improve performance.
* It seems that the new architecture achieves similar benchmark average accuracy and perplexity compared to pure Transformer. Although the authors claim that it is 10% faster than Mistral, there are many acceleration methods for pure Transformer models like GQA and kernel fusion which may mitigate the gap. What is the main advantage of the proposed architecture against Transformer?

**Questions:**

* For the hybrid model baseline, what is the architecture (activation, layer ordering, FFN size, and attention-Mamba ratio)?
* In previous work, usually hybrid models (e.g., Griffin, Jamba, Samba, Zamba) can slightly outperform pure Transformer or pure Mamba. Why does it perform worse than Transformer or Mamba in this work?
* How is the proposed model's performance on real-world retrieval tasks like phonebook lookup and needle-in-a-haystack?
* What is the impact of window length for the sliding window attention?
* How is the profiling of the "KV Cache" of the proposed method in terms of sequence length?

**Limitations:**

Please check the above.

---

> ### Author Rebuttal · Authors · 2024-08-07
>
> Thank you for your feedback and your suggestions. We've addressed the main concerns shared with other reviewers in the global comment, here we will address specific concerns.
>
> **Lack of sufficient ablation study.** Throughout our work (see Table 1, 2, and Figures 2, 3, 4, and 5), we compare B’MOJO to BMOJO-F (B’MOJO without Long-Term Eidetic Memory) in order to ablate the contribution of Long-Term Eidetic Memory, we further compare B’MOJO-F to a vanilla Hybrid Baseline (B’MOJO-F without Fading Memory) in order to ablate the contribution of Fading Memory, and we ablate Hybrid Baseline with Mamba to ablate the contribution of Short Term Eidetic Memory. The contribution of permanent structural memory is trivially ablated by the fact that trained weights outperform untrained ones. This covers all needed ablations: Overall, we find that B’MOJO > B’MOJO-F > Hybrid baseline > Mamba, ablating the role of each individual component.
>
> **Impact of short-term eidetic memory.** Short term eidetic memory in-context refers to the most recent set of tokens (sliding window) processed by the model, analogous to the context of a standard transformer model. For typical language modeling it is impactful --- it is used by the model to process and recall information from the most recent past (akin to a n-gram model). However it has no impact on long range recall capabilities beyond the context length. In both Tables 1 and 2, we refer to long-term eidetic memory, not short-term.
>
> **Adding eidetic memory does not improve performance.** Our scaling law results, see Figure 3 show that the eidetic memory has a positive impact on B’MOJO, which decreases perplexity as we scale the model size. To further isolate the eidetic memory gains we followed reviewer’s suggestions and report in Table A1 results on NVIDIA’s RULER benchmark using our larger models: B’MOJO-F 1.4B (with only fading memory) and B’MOJO 1.4B (with fading + eidetic memory). We show that B’MOJO-F is strictly weaker and the gap is higher the longer the required context length. Complementing the perplexity results, this showcases that our innovation selection mechanism indeed helps the model to recall specific information from the past. Furthermore, this is in line with our synthetic experiments in Figure 2 albeit at larger scale. Note that in Figure 2 panel 4 we show that increasing the number of eidetic memory tokens produces an increase in recall performance prior to saturation.
>
> **The main advantage of the proposed architecture** Our proposed model’s main advantage over Transformers is that it simultaneously 1) introduces linear inference complexity with respect to context length and constant KV cache while 2) preserving Transformers’ performance without incurring in the quadratic complexity cost of Attention.
>
> **Acceleration methods.** Both GQA and Kernel Fusion speed up Transformers by efficiently moving data in memory, but neither change the fundamental quadratic inference complexity of Transformers. Even for moderate context lengths these acceleration methods cannot close the linear vs quadratic gap between B’MOJO and Transformers. We make this point empirically in Table A2, where we report profiling results on both our Transformer baseline and B’MOJO with and w/o GQA at the 1.4B/3B scale for different context lengths (1k/2k/4k). Note GQA can be applied to all attention layers in B’MOJO for further acceleration. In all cases, B’MOJO is still faster than Transformers w/ and w/o GQA. GQA does help Transformers more than B’MOJO, and the relative speed gap for, e.g. at the 1.4B scale, with 2k context length case is reduced from ~10% to 7% . As expected, B’MOJO’s improves more as longer sequences are used due to the quadratic vs linear scaling of attention on the Transformer baseline.
>
> **Empirical results B’MOJO vs Mamba vs Transformer.** When performing apples-to-apples comparisons (i), our largest B’MOJO model (1.4B) outperforms Mamba 1.4B (see Table 1, Figures 3, 5 and the new experiments with the needle-in-a-haystack results in Table A1, in the global comment), a trend observed in our synthetic experiments too (see Figure 2). While prior works like Jamba (and the recently released Zamba) slightly outperform pure Transformer models they used full attention, and thus retain the quadratic dependence of Transformers. In contrast, our work only uses a small 512 token sliding window attention to produce a model with linear dependency on the sequence length with a constant KV cache size. Other works like Griffin and Samba (which has been published after the submission of this manuscript) also use sliding windows, but manage to slightly outperform Transformer leveraging much longer sliding windows sizes.
>
> **Impact of the window length and profiling of the KV Cache.** The window length measures short-term eidetic memory and is similar to the impact of changing the window size on Transformer models like Mistral, the larger the better the results but at a higher FLOPs count, all our experiments leverage a fixed window size of 512.
> B'MOJO's forward time is constant with respect to the sequence length, and its KV cache is comprised of the sliding window cache, the fading memory and eidetic memory tokens (see Figure 7 for more details), whose number is fixed by the user.
>
>
> ### Table A2: Profiling time (in ms) of a forward call for various model sizes, batch size = 1, measured on A100 40Gb
>
> |fwd profiling time in ms||Context Length|| B'MOJO's rel. improvement||
> |---|---|---|---|---|---|
> |Models|1024|2048|4096|||
> |---|---|---|---|---|---|
> |**1.4B**||||||
> |Transformer w/o GQA|72|117|oom|||
> |Transformer|67|110|oom|||
> |BMOJO-F|56|106|207|29%|10%|
> |BMOJO-F with GQA|53|99|190|26%|11%|
> |BMOJO|62|107|229|16%|9%|
> |BMOJO with GQA|58|103|224|16%|7%|
> |---|---|---|---|---|---|
> |**3B**||||||
> |Transformer w/o GQA|83|173|oom|||
> |Transformer |76|161|oom|||
> |BMOJO-F|78|157||6%|10%|
> |BMOJO-F with GQA|73|142||4%|13%|
> |BMOJO|81|163|311|2%|6%|
> |BMOJO with GQA|75|148|286|1%|8%|

---

### Author Rebuttal · Authors · 2024-08-07

We thank the reviewers for their constructive feedback and the positive comments on the novelty of our method (MJ98, Plgm, Wgie) as well as its theoretical motivation and connection with Stochastic Realization Theory (MJ98, Wgie). Furthermore, we are glad that reviewers appreciated our efficient implementation which allows us to process longer contexts “more efficiently than standard attention” (majr) and our “impressive OOD generalization results” (Plgm) outperforming the transformer baseline in length generalization (wgie).

**Main evaluation benchmarks and paragons.** Reviewers MJ98, majr and Wgie are concerned that our Transformer baseline seems to perform comparably or better than our proposed model, with the exception of the length generalization task. Our experimental results are of two main types, (i) a generic comparison on short context benchmarks typically used to assess LLMs’ zero-shot performance on short contexts, and (ii) specific long context/recall-based tasks where finite-context models are ill-fit. For a fair apples-to-apples comparison all our models are trained using the same pre-training data, tokenizer, and context length. We wish to emphasize that in this setting, and for the results of type (i), Transformers are a paragon, not a baseline, since most tasks are answerable within the context. Therefore in the results of type (ii) we leverage specific benchmarks, like synthetic tasks, long context evaluation and length generation to assess our models. The goal of our novel model class is to cover the entire spectrum, i.e. perform comparably to the paragon on finite contexts while preserving higher performance than Transformers whenever the data relevant to solve the inference tasks fall outside the context window. While the latter tasks may be a minority by frequency, they carry outsize weight in terms of business value (e.g. to support more factual queries and reduce hallucinations, RAG, …). Unlike the paragon, our baselines Mamba and a Hybrid model are surpassed by B’MOJO in both (i) and (ii).

**Better assessment of long context recall capabilities.** Reviewers MJ98 and majr suggested we further assess the longer context recall capabilities of B’MOJO on a larger and more difficult needle-in-a-haystack benchmark. Following their suggestions, we report our results of our larger models (1.4B) in Table A1 below using NVIDIA’s RULER benchmark. Our goal is to test a) how much adding eidetic memory helps complementing fading memory, b) how B’MOJO compares with a Transformer and c) how it compares with a Mamba model on longer contexts.
We test our pre-trained models on 2k tokens at varying context lengths (2k and 4k) and compare our models with Transformer and Mamba baselines pre-trained from scratch on the same data. We find:

a) B'MOJO (with fading + long term eidetic memory) is strictly stronger than BMOJO-F (with only fading memory), and the gap is higher with longer context lengths (51% vs 48% accuracy @2k and 19% vs 10% accuracy @4k respectively). This showcases that our innovation selection mechanism helps the model to recall information from the past (in line with our synthetic experiments in Figure 1).

b) The Transformer baseline only has high recall when it is tested on the same context length used during pre-training (2k) and struggles with longer sequences (as we show in Figure 5 on the length generalization experiments). On the other hand, B’MOJO can recall information beyond the pre-training context length, outperforming the Transformer baseline on 4k context length.

c) Both B’MOJO and B’MOJO-F outperform the Mamba baseline and the gap increases as the context length increases, showcasing that Mamba does not preserve enough information from the older past and this is especially evident when the context length increases (accuracy @4k is 8% for Mamba and 19% for B'MOJO).



### Table A1: Long context evaluation with RULER (needle in a haystack)
| Context Length | Model        | S-NIHA | MK-NIAH | MV-NIAH | MQ-NIAH | Average |
|----------------|--------------|--------|---------|---------|---------|---------|
| 2048           | Transformer  | 100    | 95      | 62      | 61      | 79      |
|                | Mamba        | 100    | 32      | 29      | 28      | 47      |
|                | B'MOJO-F      | 90     | 36      | 35      | 31      | 48      |
|                | B'MOJO        | 90     | 45      | 37      | 33      | 51      |
|----------------|--------------|--------|---------|---------|---------|---------|
| 4096           | Transformer  | 0      | 0       | 0       | 0       | 0       |
|                | Mamba        | 9      | 12      | 5       | 7       | 8       |
|                | B'MOJO-F      | 10     | 16      | 5       | 8       | 10       |
|                | B'MOJO        | 22     | 21      | 17      | 17      | 19      |

---

### Decision · Program_Chairs · 2024-09-25

**Decision:**

Accept (poster)

**Comment:**

The submission studies an interesting combination of recurrent architecture motifs (such as Mamba) with Transformers. Three out of four reviewers were satisfied with the authors’ responses and raised their scores. The sole negative reviewer did not participate in the discussion. However, it looks like their main concerns about not having ablation studies and discussion of acceleration methods (such as GQA) has been addressed by the authors. The ablations are already present in the paper, and I suggest that the authors make it clearer in the camera ready version. The authors should also include a more comprehensive discussion of related work on modern hybrid architectures (attention + SSM), see comments by reviewer MJ98.